# Streptolysin O accelerates the conversion of plasminogen to plasmin

Di Tang [1] ✉, Hamed Khakzad [2], Elisabeth Hjortswang[1], Lars Malmström[1], Simon Ekström [3], Lotta Happonen [1] & Johan Malmström [1,3] ✉

Group A *Streptococcus* (GAS) is a human-specific bacterial pathogen that can exploit the plasminogen-plasmin fibrinolysis system to dismantle blood clots and facilitate its spread and survival within the human host. In this study, we use affinity-enrichment mass spectrometry to decipher the host-pathogen protein-protein interaction between plasminogen and streptolysin O, a key cytolytic toxin produced by GAS. This interaction accelerates the conversion of plasminogen to plasmin by both the host tissue-type plasminogen activator and streptokinase, a bacterial plasminogen activator secreted by GAS. Integrative structural mass spectrometry analysis shows that the interaction induces local conformational shifts in plasminogen. These changes lead to the formation of a stabilised intermediate plasminogen-streptolysin O complex that becomes significantly more susceptible to proteolytic processing by plasminogen activators. Our findings reveal a conserved and moonlighting pathomechanistic function for streptolysin O that extends beyond its well-characterised cytolytic activity.

Group A *streptococcus* (GAS), or *Streptococcus pyogenes*, is a gram positive human bacterial pathogen that can cause a wide range of infections. Worldwide, GAS is responsible for 727 million superficial infections such as pyoderma and pharyngitis, 1.78 million severe infections, and over 500,000 deaths annually[1]. Infections caused by GAS can additionally lead to autoimmune-associated sequalae such as rheumatic fever/heart disease and poststreptococcal acute glomerulonephritis[2]. Recently, several studies have observed antibiotic-resistant clinical GAS isolates[3,4], which represents a substantial public health concern due to the high prevalence of GAS infections. These observations emphasise the need for more in-depth understanding of GAS pathogenesis to promote the development of innovative therapeutic strategies.

Infections caused by GAS activate the coagulation system leading to the formation of a fibrin network to entrap the bacteria[5]. As a consequence, GAS has developed the capacity to exploit the human haemostatic system to promote microbial survival and dissemination. In particular, several mechanisms to control the fibrinolysis pathway and

the plasminogen (PLG)/plasmin (PLM) system have evolved in GAS to induce fibrinolysis and to escape entrapment[6]. PLG circulates as two different glycosylated proteoforms (type I: $Asn_{308}$, $Thr_{365}$; type II: $Thr_{365}$)[7]. The mature form of PLG is a 90 kDa protein, subdivided into seven consecutive domains. The first domain is referred to as the Pan-Apple N-terminal (PAN) domain, followed by five homologous kringle domains (K1-K5) and a peptidase S1 domain (PSD). Circulating PLG adopts a tight conformation held together by intramolecular inter-domain interactions to prevent aberrant activation under normal conditions[8]. When PLG interacts with fibrin or cellular receptors, a competitive binding occurs with its lysine-binding sites, leading to a conformational change in PLG to expose the activation loop[9]. This activation loop is subsequently cleaved by plasminogen activators (PAs) to form active plasmin[10], where the two cleaved chains remain interconnected by disulphide bonds. Cleavage of the activation loop and proteolytic removal of the PAN domain leads to the formation of mature plasmin. Small-angle X-ray scattering (SAXS) techniques have revealed the conformational transition of PLG from its closed to open

[1]Division of Infection Medicine, Department of Clinical Sciences Lund, Faculty of Medicine, Lund University, Lund, Sweden. [2]Université de Lorraine, CNRS, Inria, LORIA, Nancy, France. [3]SciLifeLab, Integrated Structural Biology Platform, Structural Proteomics Unit Sweden, Lund University, Lund, Sweden. ✉e-mail: di.tang@med.lu.se; johan.malmstrom@med.lu.se

form, showing a single-step conversion process that results in multiple final conformations with high flexibility in solution[9]. However, interfering signals from macromolecular PLG-binding partners, such as bacterial proteins and cellular receptors, make SAXS ineffective in detecting these changes under more complex conditions. Currently, several PAs have been discovered, including tissue-type plasminogen activator (tPA), urokinase-type plasminogen activator (uPA), kallikrein and factor XII (Hageman factor)[11], as well as several bacterial proteins from different species[12].

So far, five streptococcal proteins have been reported to interact with PLG: PLG-binding GAS M-like proteins (PAM)[13], α-enolase (SEN)[14], glyceraldehyde 3-phosphate-dehydrogenase (GAPDH)[15,16], extracellular protein factor (Epf)[17] and streptokinase (SKA)[18]. Streptokinase is a well-characterised PLG-binding streptococcal protein that forms a 1:1 activator complex with PLG to catalyse the conversion of other free PLG molecules into active plasmin[19]. PAM binds to PLG between the K1-K2 domains using the a1a2 motif to introduce a relaxed conformation that renders PLG more susceptible to activation. The PAM-mediated increase in PLM activity can occur either through co-operation with SKA activation or directly by exploiting host plasminogen activators[20]. Notably, a similar enhanced conversion has been observed in the binding of α-Enolase to PLG[21]. Importantly, previous reports have shown that GAS virulence was enhanced in a human-PLG-transgene mouse model[22], demonstrating that the interaction between GAS and PLG is critically important for GAS pathogenicity, dissemination and survival[23].

As one of several major virulence factors, streptolysin O (SLO) is a 60 kDa pore-forming toxin secreted by most GAS strains. The first 69 N-terminal residues of SLO form a disordered region, followed by three discontinuous domains (D1-D3) and a membrane-binding domain (D4) at the C terminus[24]. SLO primarily binds to cholesterol-rich membranes and oligomerises to form a pre-pore followed by D3-driven transition into a pore to cause cytolysis[25,26]. Beyond this main biological function, SLO also translocates NAD-glycohydrolase into host cells[27], promotes streptococcal superantigen activity[28], acts as an immune-modulatory molecule for neutrophils[29,30] and impairs phagocytic clearance of GAS[31,32] and intracellular lysosomal killing[33]. Recent studies on other members of the cholesterol-dependent cytolysin (CDC) family proteins such as pneumolysin (PLY)[34] and perfringolysin O (PFO)[35] have shown that this protein family has several biological functions beyond pore-formation, like reducing inflammation or binding to glycans. Several studies also reported that there are other non-classic cholesterol CDC cellular receptors[36] and CDC-binding plasma proteins like human albumin[37,38]. SLO is present in more than 98% of the sequenced GAS genomes with a relatively low sequence variability of less than 2%[39], making it a suitable target for developing therapeutic interventions.

In this work, we first applied affinity-enrichment mass spectrometry (AE-MS) to investigate the interactome landscape formed between SLO and human plasma proteins, revealing that SLO binds specifically to PLG. In contrast to streptokinase, the binding of SLO to PLG does not result in direct activation of PLG but renders PLG more sensitive to tPA and SKA activation. Using an integrative structural mass spectrometry strategy combining hydrogen/deuterium mass spectrometry (HDX-MS), cross-linking mass spectrometry (XL-MS) and computational modelling[40], we demonstrate that SLO binding to PLG leads to a stabilised intermediate complex, which facilitates PAs in catalysing increased plasmin production.

## Results

### A protein-protein interaction network between SLO and human plasma proteins

We have previously shown that many virulence factors produced by GAS form protein networks with human plasma and saliva proteins[41].

To further expand these interaction networks, we produced two virulence factors from GAS in recombinant form, SLO and C5a peptidase (SCPA), genetically fused to an affinity tag. The tagged bait proteins were used to enrich interacting human proteins from pooled healthy plasma, which were identified by high-resolution LC-MS/MS (AE-MS) (Supplementary Fig. 1a) using tagged green fluorescence protein (GFP) as a reference control throughout experiments. The tagged GFP bait was used to filter out non-specific protein interactions, determine the baseline background noise and facilitate comparison with previous studies[41,42]. The two GAS proteins enriched distinct sets of human plasma proteins as shown in the clustered heatmap and PCA plot (Supplementary Fig. 2a, b). These protein sets were organised into a bait-prey matrix and processed using the MiST workflow[43,44]. The resulting network was visualised using Cytoscape[44,45] to demonstrate the 90 plasma proteins interacting with the two GAS bait proteins, forming 116 bait-prey edges (Supplementary Fig. 3a). Notably, 42 of these interactions were exclusively formed with SLO. Integrating the SLO-centric network with the high-confidence interactions from the STRING database[46] expanded the network to a total of 203 edges, revealing subnetworks predominantly composed of other plasma proteins, complement-related proteins and immunoglobulins (Fig. 1a).

Functional over-representation analysis of the SLO enriched plasma proteins revealed a network of terms such as immunoglobulin-associated functions, regulation of complement activation and negative regulation of blood coagulation (Fig. 1b, Supplementary Fig. 4a, b). A prominent feature of the interaction networks was the enrichment of several immunoglobulin G (IgG) chains (Supplementary Table 1) that were specifically enriched by SLO or SCPA, alongside components of the complement system that are known to interact with the fragment crystalisable (Fc) part of IgG, such as C1q. This observation shows that the pooled healthy human plasma contains multiple IgG clones specific for both SLO and SCPA, which was corroborated using enzyme-linked immunosorbent assay (ELISA) (Supplementary Fig. 3b).

In the next step, we used FDR-adjusted multiple t-tests comparing SLO/GFP and SLO/SCPA to identify significant protein-protein interactions specific for SLO using thresholds of fold change >1.0 and adjusted $p$-value of <0.01 (Fig. 1c, d). The differential analyses consistently identified several prey proteins including apolipoprotein E (APOE), plasminogen (PLG), clusterin (CLU), vitronectin and plasma protease C1 inhibitor (SERPING1) as the most significant interactions that were specific for SLO. It is noteworthy that these interactions were formed even in the presence of anti-SLO IgGs, suggesting that SLO-specific IgG does not outcompete these interactions.

To further pinpoint the most relevant SLO-binders and to imitate the microenvironment formed during plasma leakage[47], the pooled human plasma was diluted (50% and 10% plasma) and the AE-MS experiments were repeated three times per dilution. The SLO-specific interactors identified above were quantitatively monitored in the pulldown samples from all three baits (SLO, SCPA and GFP) across the three plasma dilutions. Consistent across the undiluted and the 50% diluted conditions, PLG, APOE and CLU were significantly enriched by SLO when compared to SCPA and GFP separately (Supplementary Fig. 5a, b). On the other hand, only PLG was determined to be significantly enriched in all significance tests using the 10% diluted plasma samples (Fig. 2a, b). In conclusion, we used AE-MS to map two plasma protein interaction networks formed around SLO or SCPA and suggest several binding partners, including plasminogen, that specifically interacted with SLO.

### SLO binds specifically to PLG and enhances both tPA-mediated and SKA-mediated conversion of plasminogen to plasmin

Zymogen plasminogen circulates under normal conditions in a tight conformation resistant to proteolytic activation (Supplementary Fig. 6a) to prevent the formation of PLM[48]. However, upon binding to

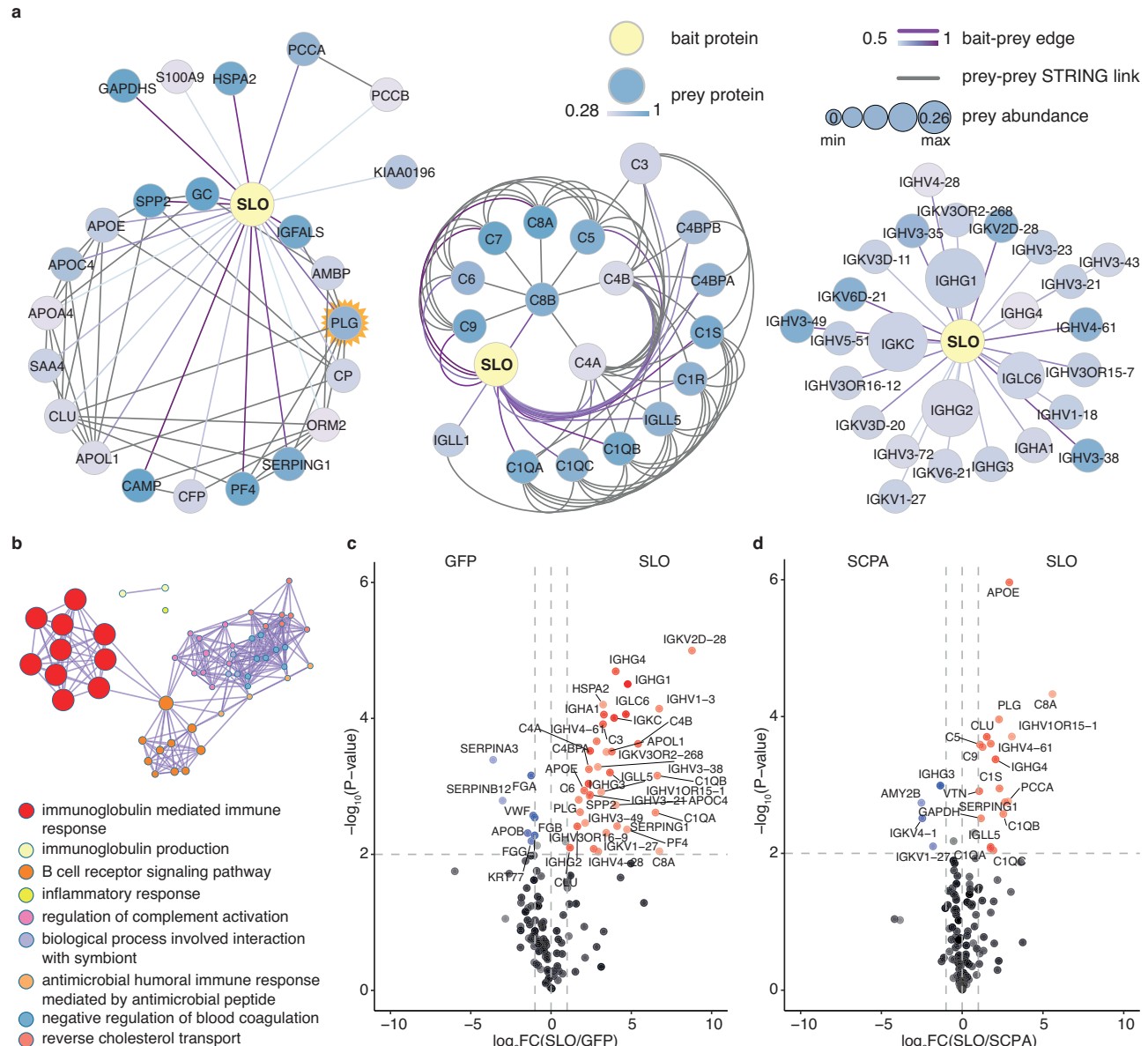

**Fig. 1 | A protein-protein interaction network between SLO and human plasma proteins.** Recombinant tagged SLO (streptolysin O) and SCPA (C5a peptidase) baits were used to enrich proteins from undiluted pooled human plasma in three independent experiments, followed by DIA (data independent acquisition) mass spectrometry analysis. The affinity-enrichment DIA quantification data was sorted into a bait-prey matrix and scored by MiST (Mass Spectrometry interaction STatistics) and visualised in Cytoscape. **a** Interaction subnetworks between SLO and plasma proteins. Nodes represent proteins, with node size and colour indicating abundance and specificity metrics of the enriched prey protein, while the connecting edges represent protein-protein interactions (PPIs). Edge colour reflects the MiST score value. The SLO-plasma PPI network was further integrated with high-confidence PPI from the STRING interactome database, identifying associations among prey proteins. Subnetworks were generated by manual curation: left, network of other plasma proteins; middle, network of complement-related proteins;

right: network of immunoglobulins, (**b**) A clustered network comprising GO:BP (Gene Ontology Biological Process) enriched terms extracted from the enrichment analysis querying the 68 interacting prey plasma proteins. The most representative term from the top 9 GO:BP clusters are shown in the legend. Volcano plots illustrate statistical protein abundance comparisons between (**c**) SLO and GFP, and (**d**) SLO and SCPA, with calculated FC (fold change) plotted against $-\log_{10}$(P-value) from FDR-corrected multiple t-tests (two-sided) on extracted ion intensities. Dashed lines define the significance thresholds of |fold change| > 1 and adjusted P-value < 0.01. Proteins surpassing this threshold in the SLO comparison are marked as red dots, those in the GFP/SCPA comparison in blue and non-significant proteins are coloured black. All significantly enriched prey proteins are labelled with gene names, with the colour intensity of the dot representing the corresponding prey abundance. Source data are provided as a Source Data file.

fibrin or cell surfaces, PLG shifts to a more relaxed conformation, enabling rapid conversion to active plasmin (PLM) by the plasminogen activators[10]. Because PLM shares the same primary structure with PLG and are indistinguishable by bottom-up MS analysis, we first employed an indirect ELISA assay to determine if SLO binds to both proteins. The wells were coated with equal amounts of either PLG or PLM, followed by incubation with SLO. The level of SLO binding was then quantified

using an anti-SLO monoclonal antibody and an HRP-conjugated secondary antibody. The results revealed that SLO binding to PLG is significantly higher compared to PLM (Supplementary Fig. 6b). To determine if SLO can bind to both main isoforms of human plasminogen (type I: $Asn_{308}$, $Thr_{365}$; type II: $Thr_{365}$), targeted glycoproteomics was used to search for glycopeptides in the AE-MS DDA datasets. The identification of glycopeptides containing N-linked $Asn_{308}$

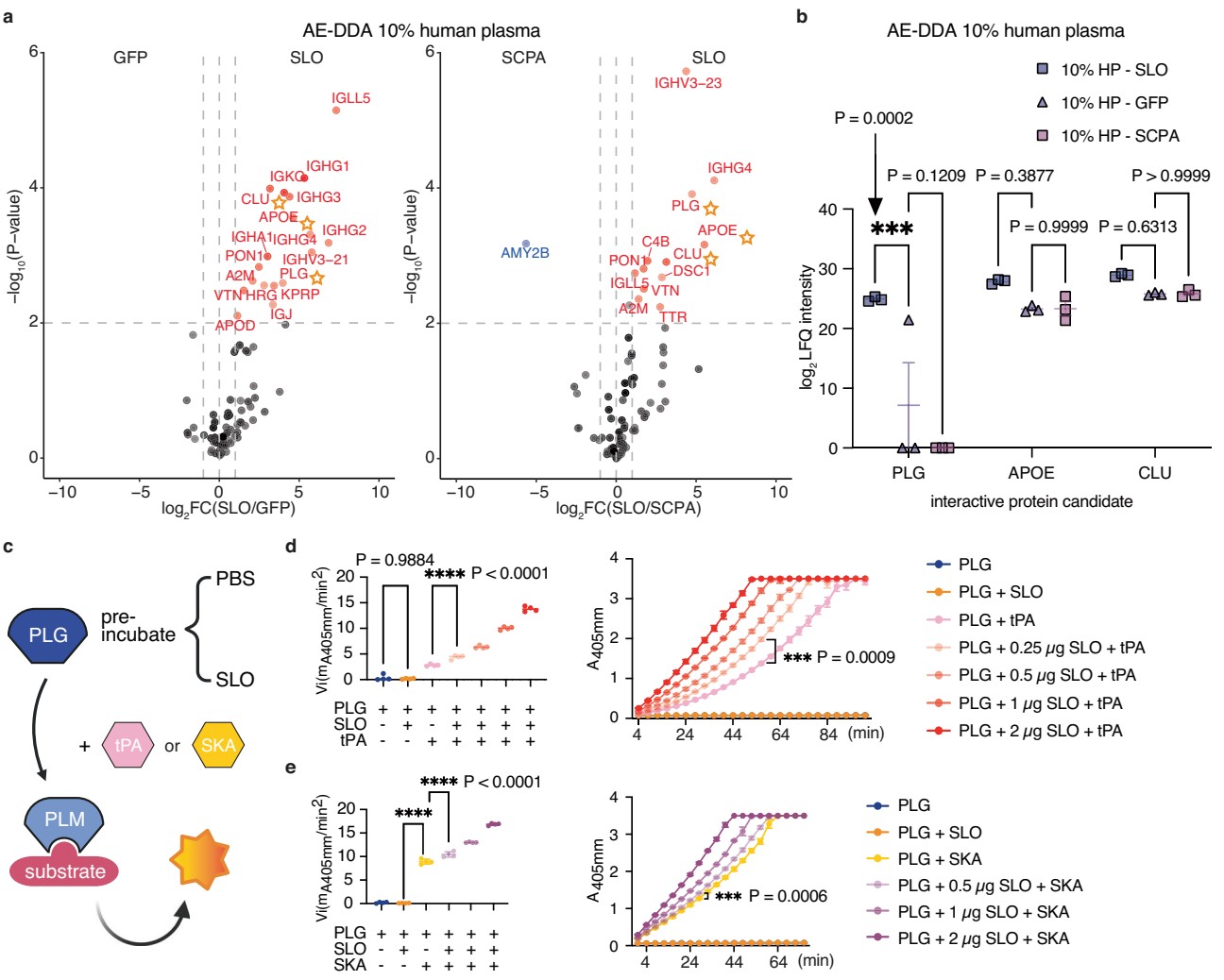

**Fig. 2 | SLO binds specifically to PLG and enhances both tPA- and SKA-mediated conversion of plasminogen to plasmin.** The pooled human plasma was further diluted and analysed using AE-MS in three independent experiments to identify proteins that interact specifically with streptolysin O compared to the GFP/SCPA as reference controls. **a** Differentially SLO-enriched protein was determined by comparing normalised DDA (data dependent acquisition) LFQ (label-free quantification) intensities. Here, 10% diluted plasma was used as prey mixture. The same statistical test, significance thresholds and colour scheme were applied as in Fig. 1c, d. Plasma proteins determined to be significant across all dilution condition in both comparison are marked with star symbols. **b** Two-way ANOVA analysis followed by Tukey's post-hoc test (two-sided) on the log$_2$-transformed LFQ intensities of the selected three SLO-interacting plasma protein candidates (marked with stars in (**a**)). **c** Schematic outline of the plasminogen activation assay. The assay involves pre-incubating PLG without/with different amount of SLO, followed by the

addition of tissue-type plasminogen activator (tPA) or streptokinase (SKA), and a chromogenic PLM substrate. The absorbance was continuously measured and reflects the amount of plasmin that was generated correlating to PLG activation rate by the corresponding activator. PLG activation assay using (**d**) tPA or (**e**) SKA as activator. Right: A comparison of the initial PLG activation velocity (Vi) was made using one-way ANOVA followed by Tukey's or Dunnett's post-hoc test (two-sided) across all conditions. +: with; -: without. Left: connected dot plot shows the cumulative effect of PLG activation over time and tested for statistical significance using a two-way ANOVA followed by Dunnett's post-hoc test (two-sided). Significance levels are indicated as *, **, ***, and ****, corresponding to adjusted *P*-values of <0.1, <0.01, <0.001 and <0.0001, respectively. Either individual dot or super-posed symbol at mean is plotted, with standard deviation as error bar. **c** created with Biorender.com. Source data are provided as a Source Data file.

(Supplementary Fig. 6c) and O-linked Thr$_{365}$ in all SLO-enriched plasma pulldowns indicates that SLO can bind to both predominant proteoforms of PLG.

Unlike streptokinase, our initial experiments showed that SLO does not directly catalyse the conversion of PLG to PLM. Consequently, we next investigated whether binding of SLO could alter PLG sensitivity to tPA activation in a similar fashion to fibrin or cell surfaces. In these experiments, PLG was incubated with SLO prior to the addition of tPA and a chromogenic substrate for PLM (Fig. 2c). The initial activation rates were calculated and shown as individual data points for four replicates across multiple conditions. Additionally, kinetic mode analysis was conducted to monitor the accumulation of the

end-product (chromophore para-nitroaniline) throughout the incubation period, until the positive control (PLG + tPA) reached saturation. tPA activation of PLG was greatly enhanced with the presence of SLO (Fig. 2d). Next, we also investigated the role of SLO binding in SKA activation. Similar to tPA activation, SLO significantly accelerated the SKA-mediated conversion of PLG to PLM (Fig. 2e). These results show that SLO can enhance both the tPA-mediated and SKA-mediated conversion in a dose-dependent manner without affecting the activity of PLM (Supplementary Fig. 6d). Furthermore, we determined that SLO binding to PLG does not affect the haemolytic function of SLO, suggesting that these two interactions can co-occur and are not mutually exclusive (Supplementary Fig. 6e).

## HDX-MS reveals protected regions and protein dynamics of PLG upon SLO binding

To map the binding interfaces between PLG and SLO and to investigate the dynamic aspects of the interaction, we performed HDX-MS to explore changes in deuterium uptake in PLG upon binding to SLO (Supplementary Fig. 1b). In HDX-MS, changes in deuterium uptake indicate interaction interfaces or alterations in protein state/conformation during processes such as protein-protein binding[49] or protein-small molecule drug interactions[50]. Here, PLG was analysed in both the apo (unbound) and the SLO-bound states, where HDX-MS identified 150 common high confidence peptides, achieving 65.3% sequence coverage with many overlapping peptides (Supplementary Fig. 7a). Serial changes in deuterium uptake (ΔDU) across five deuteration intervals (0, 30, 300, 3000 and 9000 s) for each peptide were summed and visualised in a butterfly plot (Supplementary Fig. 7b).

In our first analysis, PLG peptides with significantly decreased/increased deuterium uptake were projected onto the PLG structure in surface presentation (Fig. 3a), indicating a global stabilisation of PLG over time. The same significantly determined peptides shown in the Woods plot in Fig. 3b pinpoint several protected regions in PLG upon SLO binding.

In the next step, we performed a more rigorous hybrid significance test to identify the most relevant protected regions. In this analysis, overlapping peptides corresponding to the regions 34–48 and 80–88 of PAN, 405–421 in K4 and 726–734 in PSD were significantly protected within deuteration times of ≤300 s, indicating that these regions are involved in the initial binding of SLO (Supplementary Fig. 7c)[49]. During extended labelling time (≥9000 s), the most important regions were 192–203 and 277–288 in K2, 658–669 in PSD (Fig. 3c), which strongly suggest that these two domains of PLG are involved in the formation of PLG-SLO protein complex. Kinetic plots covering these aforementioned regions show differential changes in deuterium uptake throughout the different labelling times (Supplementary Fig. 7d and Fig. 3d). Full barcode plots showing mean/differential deuterium uptake for both apo and SLO-bound states are shown (Supplementary Data Fig. 7e).

In our HDX-MS experiments, the PAN and PSD domains exhibited the highest sequence coverage, which enabled further analysis (Fig. 3e). In this assessment, we observed increasing protection in residues 650–670 of the PSD domain. In contrast, the adjacent regions 720–730 and 795–810 in PSD, as well as regions 35–50 and 75–90 in PAN, demonstrated decreasing protection over time, possibly due to allosteric effects. The regions 620–630 and 770–790 in PSD were consistently protected across all labelling intervals, which indicates a more direct involvement of these regions for the binding of SLO. However, based on HDX data alone, we could not confidently distinguish between SLO-binding interfaces and effects from global domain stabilisation.

In conclusion, the HDX-MS data demonstrate that there is a direct interaction between SLO and PLG, suggest a stabilised stage of SLO-bound PLG and pinpoint protected regions in the PLG due to SLO binding or allosteric effects. Importantly, the absence of apparent deprotection regions indicates that PLG does not undergo major relaxation or conformational changes when bound to SLO.

## XL-MS defined interaction sites and protein dynamics of the PLG-SLO interaction

We speculated that the increased sensitivity to tPA activation could be linked to local conformational shifts in PLG during formation of the intermediate PLG-SLO complex. To further elucidate the binding interfaces within PLG-SLO complex and to assess the protein dynamics of PLG and SLO, we performed in-solution cross-linking mass spectrometry (XL-MS) (Supplementary Fig. 1c). Both disuccinimidyl suberate (DSS) and disuccinimidyl glutarate (DSG) were used to cross-link either PLG or PLM to SLO. In addition, DSS was used to (i) cross-link PLG with tPA and to (ii) cross-link PLG, tPA and SLO together to investigate possible transient heterodimeric or heterotrimeric complex formation. DSG has a spacer arm length of 7.7 Å compared to 11.4 Å for DSS, which generates shorter distance constraints between the cross-linkable reactive lysine residues. From four independent PLG-SLO cross-linked datasets, we identified 53 DSS and 78 DSG interprotein cross-linked spectrum matches (CSMs) with pLink 2, and 75 DSS and 69 DSG interprotein CSMs with MaxLynx (Supplementary Table 2). The different cross-linkers and search engines identified largely overlapping interprotein cross-linked sites, as shown in circular plots (Supplementary Fig. 8a).

The pLink 2-searched results were used to create linkage maps for generating fewer cross-link identifications that were only supported by a single cross-linked spectrum match. These linkage maps show the 16 DSS and 23 DSG cross-linked sites identified between PLG and SLO (Fig. 4a, b). Additionally, one cross-linked site was identified between PLM and SLO, supported by only one CSM. These results corroborate the ELISA result and show that SLO does not bind to PLM to the same extent as to PLG. In addition, we identified 5 inter-links (hetero-interprotein cross-link) between tPA and PLG possibly due to the slow kinetics of the activation in solution, and 12 inter-links within PLG-SLO-tPA indicating the formation of an intermediate heterotrimeric complex to facilitate tPA-mediated cleavage of PLG (Supplementary Fig. 8b).

Most of the interprotein cross-linked sites between PLG and SLO were identified in the three primary Pfam-annotated domains in PLG (Lys$_{81}$ in PAN, Lys$_{433}$ in K4 domain, Lys$_{664}$ and Lys$_{727}$ in PSD) (Supplementary Fig. 8c, d). Reciprocally for SLO, most of the inter-links were found in the domains 1, 3 and 4 (see Supplementary Fig. 9a–c, d–f for representative MS/MS spectra of DSS or DSG cross-linked peptide pairs). For the intermediate hetero-dimeric/trimeric complex, we identified inter-links between the EGF-like/peptidase S1 domain in tPA and the PAN/K4/PSD domains in PLG, as well as between EGF-like/kringle 2 domain in tPA and domain 4 in SLO when all three proteins were cross-linked together. Representative inter-links connecting the tPA EGF-like domain Lys$_{84}$ and PLG PSD Lys$_{727}$ and the tPA EGF-like domain Lys$_{84}$ and SLO domain 4 Lys$_{540}$ are shown in Supplementary Fig. 9g, h.

To investigate changes in the protein conformation/dynamics in PLG and SLO during binding, we next examined the intraprotein cross-links (intra-links) formed within PLG, PLM and SLO. These non-overlapping-associated intra-links were mapped onto the reference static structures (PLG, PDB: 4DUR; SLO, PDB: 4HSC) previously solved by X-ray crystallography. The measured distances between the backbone Cα atoms according to the unique cross-linked sites, together with corresponding CSM counts, are summarised in bar plots and circos plots presented in the Supplementary Fig. 10a–d. Overlaid density plots of the distance measurements for the mapped intra-links demonstrate that the fraction of intra-links with excessive over-length Cα-Cα distances (>60 Å) is considerably higher in SLO (12.3% of DSS intra-links; 17.2% of DSG intra-links) compared to the fraction of those found within PLG (3.8% of DSS intra-links; 3.2% of DSG intra-links) (Fig. 4c). As there have been no reports regarding major conformational rearrangement of SLO in its solution phase, these results suggest that SLO can adopt a dynamic conformation in solution when bound to PLG.

When comparing the number of intra-links in PLG to those within PLM, we observed a notable reduction of intra-links within PLM. The total number of intra-links was reduced from 192 to 76 corresponding to a reduction in CSM counts from 1177 to 322, most likely due to relaxation of the PLG/PLM after tPA cleavage (Fig. 4d). The loss of intra-links, which only form when neighbouring domains are in close proximity, suggests greater mobility of the PSD domain within PLM compared to PLG.

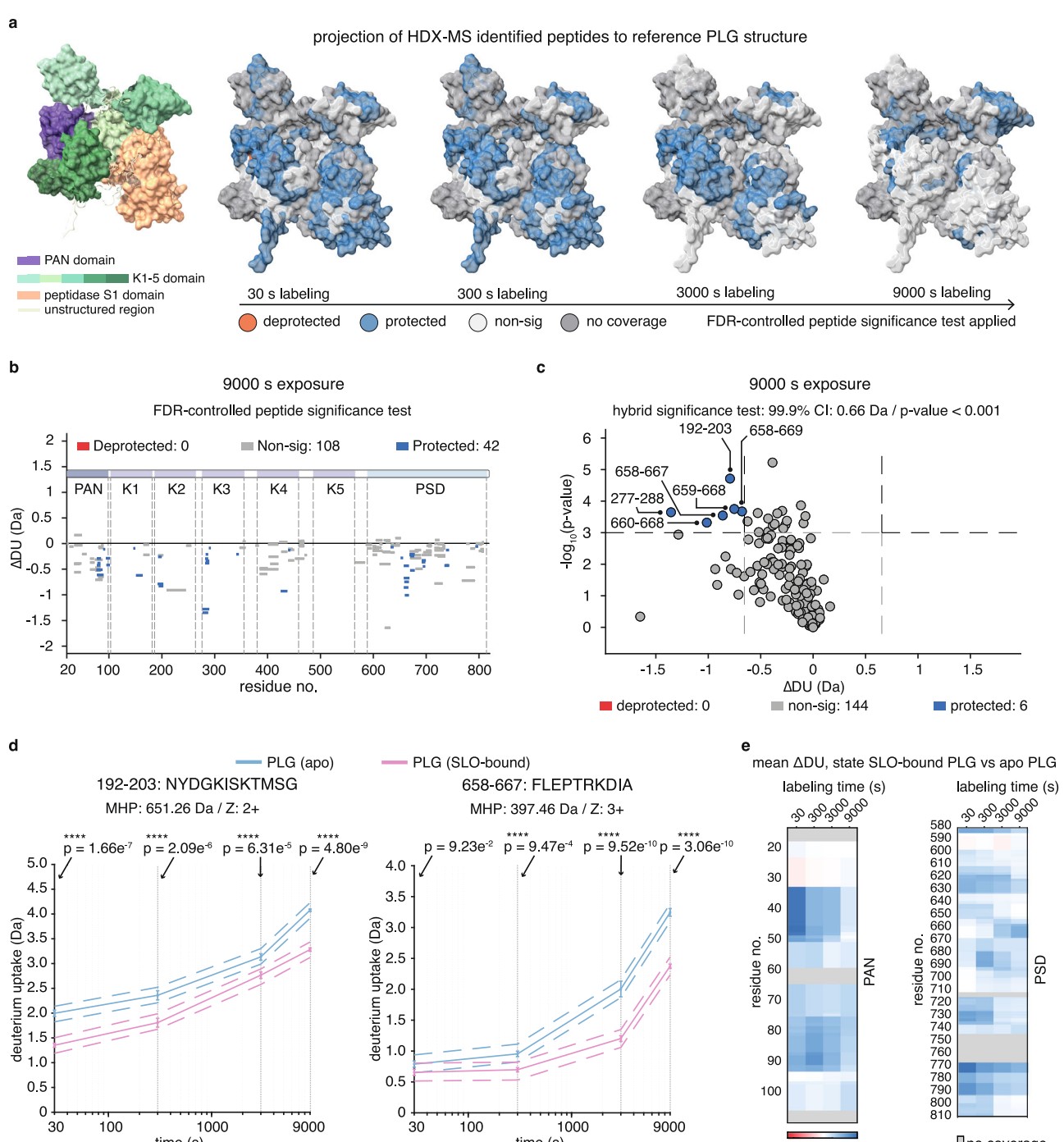

**Fig. 3 | HDX-MS reveals protected regions and protein dynamics of PLG upon SLO binding. a** Left: annotation of the seven domains in PLG shown in surface presentation. Right: projection of significantly protected/deprotected HDX-MS identified peptides onto the reference structure of PLG. Deprotected, protected and non-significance peptides was determined by peptide-level FDR-controlled significance test using a confidence level of 99.9%, across the different labelling intervals. **b** The Woods plot illustrates the deuterium uptake change of each identified PLG peptide after 9000 s deuteration, where the length of each line corresponds to peptide length, mapped onto the PLG sequence. The same FDR-corrected significance test was applied. Significantly protected peptides are shown in blue, deprotected in red, and non-significant changes in grey. **c** A volcano plot highlighting protection and deprotection after 9000 s deuteration. Dashed lines indicate the thresholds of |Da change| > 0.66 and $P$-values < 0.001 determined by a global-level hybrid significance test (two-sided). Each peptide is depicted as a dot and coloured accordingly. **d** Kinetic plots for two representative protected peptides as 192–203 and 658–667, with statistical significance determined using multiple regression analysis (two-sided) across four labelling times, colour-coded by the different states (apo and SLO-bound). Significance levels ns for non-significant and **** for >99.9% significance are marked, with Benjamini–Hochberg adjusted $P$-values. MHP refers to theoretical molecular weight of the peptide, and Z for charge state. **e** The barcode plot illustrates the mean differential change in deuterium uptake for PAN and PSD (peptidase S1) domain residues when comparing SLO-bound state to apo (unbound) state. Each residue is represented with a colour gradient indicating the extent of change as a function of labelling time. Both apo and SLO-bound state samples are prepared and measured in three independent experiments for each labelling interval. Source data are provided as a Source Data file.

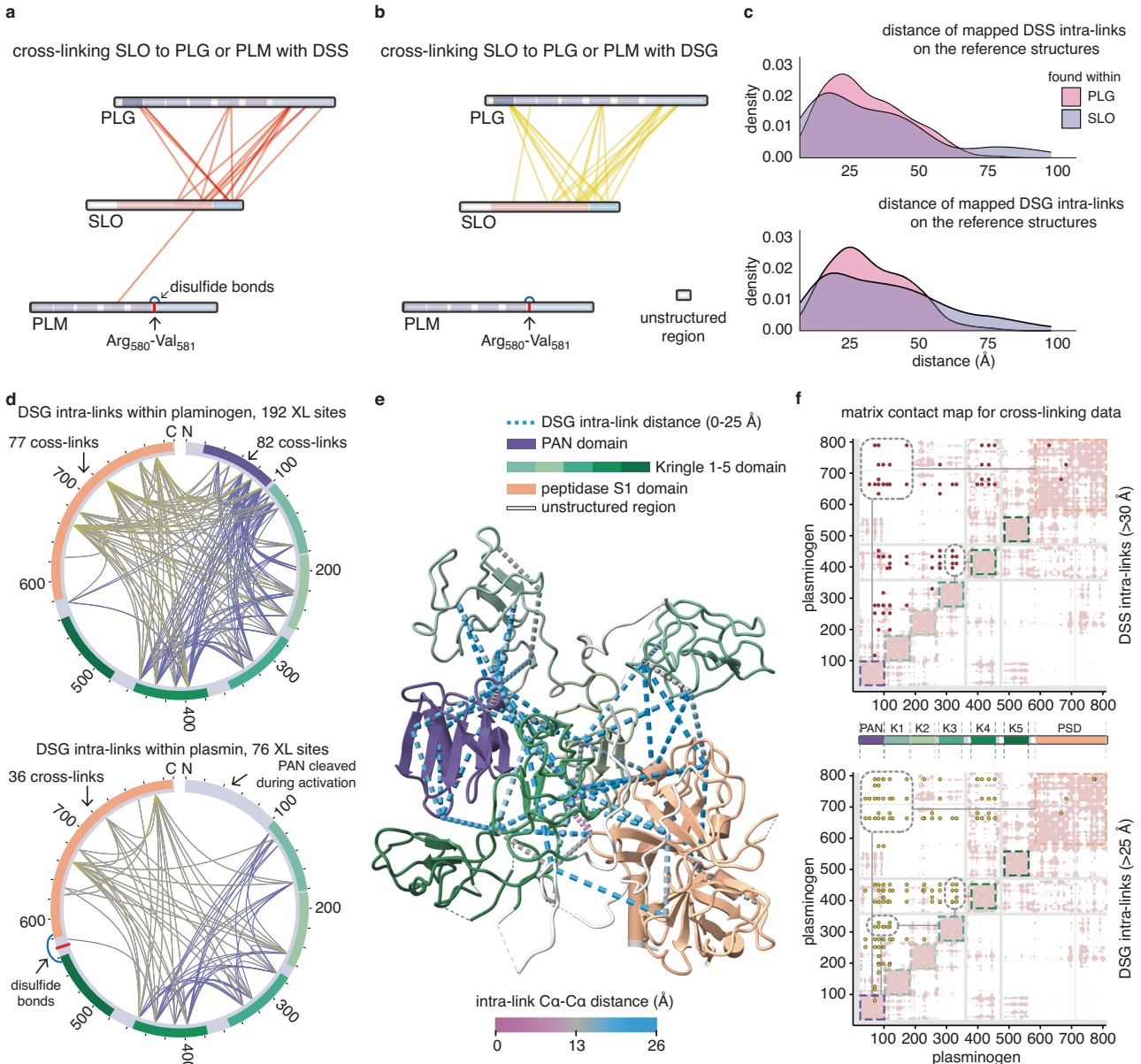

**Fig. 4 | XL-MS defines interaction sites and protein dynamics of the PLG-SLO interaction.** Linkage maps were constructed using Cytoscape based on the identified interprotein cross-links that passed the 1% false discovery rate (FDR) threshold. The cross-linked sites between SLO and PLG/PLM, with (**a**) DSS interlinks in red and (**b**) DSG inter-links in yellow, and protein family domains coloured in segments. The cleavage site involved in the conversion of PLG to PLM is depicted as a red segment, as well as disulphide bonds connecting two chains after the cut. **c** Overlaid density plots display the measured Cα-Cα distance distribution of all mapped DSS and DSG intraprotein cross-links found within PLG or SLO. **d** Two circular plots show the unique DSG intraprotein cross-linked sites as lines within PLG or PLM, with cross-links connecting to the PSD highlighted in yellow shading.

**e** A reference structure of PLG (PDB: 4DUR), coloured by domain, with DSG intra-links (Cα-Cα ≤ 25 Å) found within SLO-bound PLG shown as dot-line style pseudo bonds. **f** Two matrix contact maps for PLG showing only the distance-violating intra-links as dots (Cα-Cα > 30 Å for DSS; Cα-Cα > 25 Å for DSG). The protein sequence of PLG is annotated along both axes. A white background denotes resolved residues in the input structure, whereas a grey background indicates unresolved regions with missing structural data. The pink areas highlight residues that are in close proximity with a Cα-Cα distance ranging from 0 to 25 Å. Regions with clusters of over-length cross-links are framed with dashed line. Source data are provided as a Source Data file.

Mapping the PLG intra-links within 0-25 Å distance measurement to the reference static crystal structure, together with HDX data, suggests that the PLG conformation remains largely unchanged without any major structural rearrangement upon SLO binding (Fig. 4e). However, 75 DSS intra-links and 129 DSG intra-links still violated the allowed maximum distance in the PLG crystal structure. Plotting these distance-violating intra-links in a matrix contact map revealed that they primarily cluster in the PAN, PSD and K4 domains (dashed frames in Fig. 4f). This pattern indicates that there is movement in these

domains when PLG interacts with SLO. A similar mapping and matrix contact map on SLO suggests that domain 4 was particularly subjected to increased movement when SLO binds to PLG (Supplementary Fig. 11a–d). Taken together, these findings indicate that PLG undergoes local conformational shifts mainly driven by PAN, K4 and PSD domains during the interaction with SLO. Furthermore, the SLO-interacting proximal sites on PLG ($Lys_{81}$, $Lys_{664}$, $Lys_{727}$ and $Lys_{789}$) identified by XL-MS overlapped with the protected regions in SLO-bound PLG determined by HDX (75-90 in PAN; 650-670, 720-730 and 770-790 in PSD).

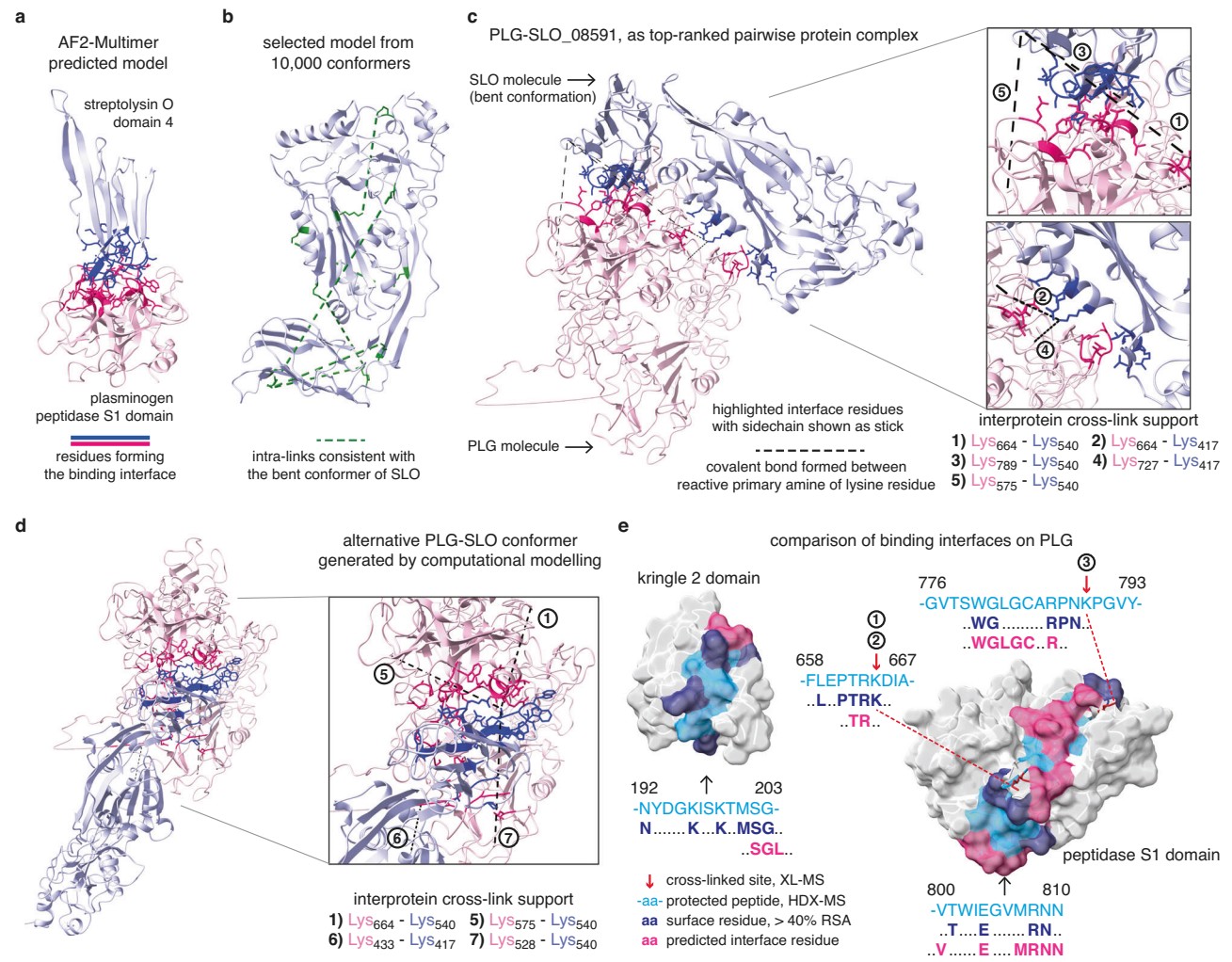

**Fig. 5 | Computational modelling predicts PLG-SLO pairwise model and illustrates the binding interfaces. a** An AlphaFold-Multimer predicted pairwise model of the SLO domain 4 and the PSD domain in PLG. The residues involved in the binding interface were identified and highlighted in different colours and shown in stick representation for the side chains. **b** The bent conformer of SLO protein was modelled by normal mode simulations. The bent model accommodates intra-links which were inconsistent with the SLO crystal structure (PDB: 4HSC). **c** The top-ranked pairwise model as PLG-SLO_08591 is presented, with the two binding interfaces highlighted in darker colours. The upper one between the PLG PSD and SLO domain 4 is supported by three cross-links, while the lower one between PLG PSD/K2 and SLO domain 3 is supported by another two cross-links. DSS linker-

formed covalent bonds are visualised as dashed lines, connecting the reactive lysine primary amines with residue numbers annotated in the legend. **d** A different set of four inter-links suggests an alternative pairwise PLG-SLO complex conformation. **e** Surface presentation and alignment comparison of interface residues within PLG-SLO protein complex and four representative HDX-MS derived protected peptides. The interface residues derived from predicted models align with the surface residues indicated in HDX-MS filtered by a relative solvent accessibility (RSA) value of > 40%. Interprotein cross-linked residues are marked by red arrows and labelled with the experimental support. Source data are provided as a Source Data file.

## Computational modelling predicts PLG-SLO pairwise model and illustrates the binding interfaces

In the final step, we performed computational modelling coupled with MS/MS analysis using previously published protocols[40] along with the identified cross-link distance constraints to determine the most important residues involved in the binding interfaces between PLG and SLO. Pairwise models of the protein complex were generated using the Rosetta docking protocol[51] that considers how well each model fulfils the inter-links between the two interacting proteins. Initial tests using AlphaFold2-Multimer[52] without any XL- or HDX-derived distance constraints resulted in a protein domain complex shown in Fig. 5a. According to this model, the membrane binding motif of SLO domain 4 binds to the PSD of PLG. However, the AlphaFold model could not explain the inter-links detected between PLG and other SLO regions like domain 3. To clarify this, we first investigated the intramolecular cross-links of SLO shown in Supplementary Fig. 10. Interestingly, we

observed multiple intra-links that were inconsistent with both the monomeric and the dimeric form of SLO, suggesting that SLO undergoes a conformational change. Computational modelling of these changes was conducted using normal mode-based geometric simulations by NMSim[53]. This analysis indicates that the domain 4 of SLO moves inward to form a bent conformation when bound to PLG that accommodates the previously over-length intra-links (K145-K540, K189-K407, K491-K540, K154-K417, K196-K298, K407-K540 and K407-K491) (Fig. 5b).

To provide additional insights into the PLG-SLO complex, two sets of 10,000 docking models were made using either the reference structure of SLO (PDB: 4HSC) or the predicted bent conformer of SLO (Fig. 5b) together with the reference PLG structure (PDB: 4DUR). Here, we selected the top model (No.08591) with the best docking energy score that was consistent with the highest number of intermolecular cross-links (Supplementary Fig. 12a) to build the 1:1

complex. In this top-ranked complex, the SLO binding sites are found in the K2 and PSD domains of PLG (Fig. 5c). Interestingly, the binding interface predicted by AlphaFold2-Multimer using domains alone closely matched the interface identified by XL-MS computational modelling using the two full-length proteins (Supplementary Fig.12b). Still, the top-ranked model was unable to explain all identified inter-links. Therefore, we cannot exclude the possibility that alternative conformers of the PLG-SLO complex may exist (Fig. 5d, Supplementary Table 3), or if two SLO molecules can bind to PLG either simultaneously or sequentially.

The HDX-MS results described above identified binding interfaces at a peptide-level resolution. In our final analysis, we calculated the relative solvent accessibility (RSA) of each residue in the native PLG structure to determine the contact residues that overlapped between HDX-MS data and the XL-MS computational modelling. For the top-ranked model obtained by the XL-MS computational modelling, six out of seven predicted interface motifs were corroborated by the calculated surface-accessible contact residues derived from the HDX results. Figure 5e presents representative alignments and comparisons, with the identified interface residues coloured based on the experimental evidence. Intriguingly, the protected peptide 192-203 (identified by HDX-MS, also shown in Fig. 3d) within kringle 2 domain of PLG, for which there were no supporting inter-links, was also identified by the computational modelling. In summary, our combined modelling revealed that SLO adopts a bent conformation that enabled the identification of the most important contact residues within the K2 and PSD domains in PLG. We conclude that leveraging both HDX-MS and XL-MS provides independent evidence that increases the confidence in the identified protein interaction interfaces.

## Highly conserved PLG-binding motifs reveal a moonlighting pathomechanism of SLO

We have previously shown that SLO is produced by >98% of the sequenced GAS genomes with a relatively low sequence variability[54]. Analysis of the interfaces required for binding demonstrates that the three PLG-binding motifs in SLO are close to 100% conserved (Supplementary Fig. 12c). These results suggest that the capacity to exploit the plasminogen-plasmin system in both streptokinase-dependent and streptokinase-independent manners is conserved in most GAS strains[55]. Further studies focusing on the low-frequency mutations and polymorphisms within the binding interface motifs are required to determine the impact of these changes on the interaction between PLG and SLO.

## Discussion

Here, we demonstrate that SLO, a well-characterised pore-forming toxin secreted by GAS[31], binds and alters the conformation of plasminogen (PLG). This interaction makes PLG more sensitive to proteolytic processing by both endogenous and exogenous activators (Fig. 6a). Binding of SLO to PLG generates a stabilised intermediate state characterised by local conformational shifts in selected PLG domains. These conformational changes ensure that host tPA and SKA gain improved access to the cleavage site in PLG, thereby accelerating the formation of plasmin (PLM) through the formation of a heterotrimeric PLG-SLO-PA (tPA/SKA) complex (Supplementary Fig. 12d, e). The cleavage of PLG by tPA induces a major conformational change in the PSD domain during the conversion, leading to detachment of SLO from mature PLM. Free plasmin then acts to degrade fibrin and dissolve blood clots, while SLO potentially reinitiates the cycle by targeting other inactivated PLG molecules (Fig. 6b). In vivo, we speculate

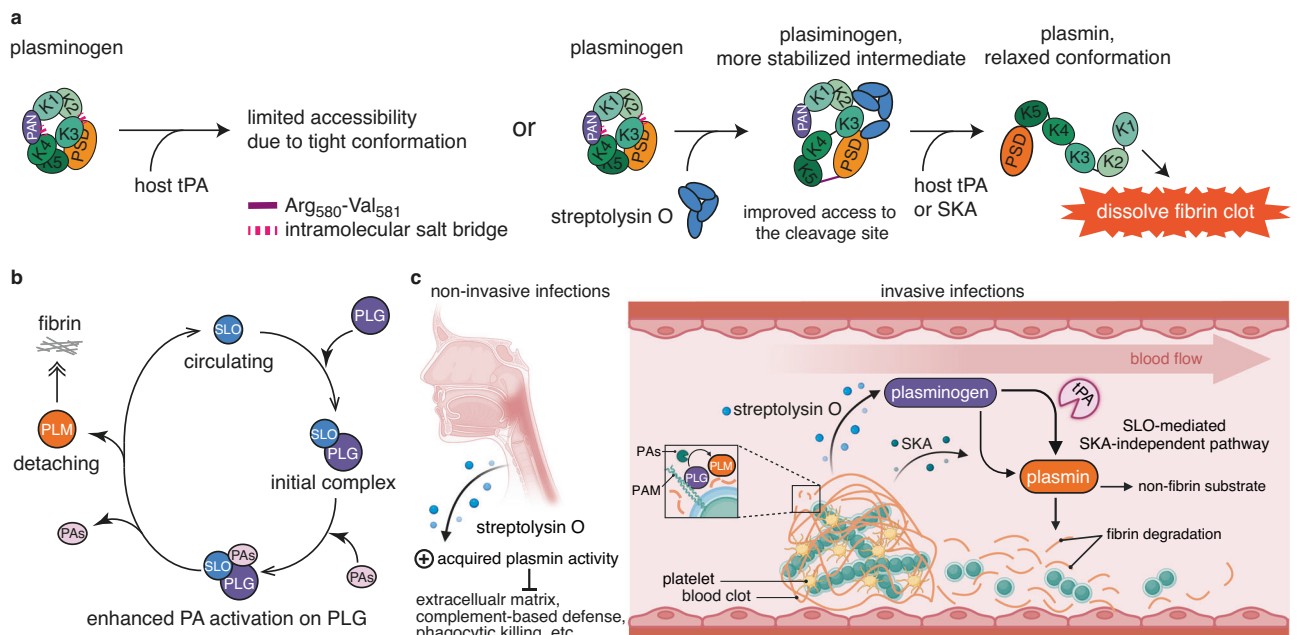

**Fig. 6 | Schematic of PLG-SLO interaction and its implication on *S. pyogenes* pathogenesis. a** PLG, depicted in a tight conformation, resists activation to prevent unwarranted enzymatic activity in solution. Interaction with the GAS-secreted toxin SLO, trigger local conformation shifts and stabilises the PLG molecule. This PLG-SLO intermediate protein complex then becomes more susceptible to activation by tissue plasminogen activator (tPA) and streptokinase (SKA), resulting in increased production of plasmin. The increase in plasmin activity leads to the degradation of the fibrin and extracellular matrix, facilitating breakdown of blood clots. **b** This pathogenic mechanism potentially allows *S. pyogenes* to further exploit the host's fibrinolysis system, promoting deeper tissue invasion and systemic dissemination by utilising and accumulating the acquired plasmin activity. **c** A proposed model of how SLO enhances both tPA and SKA activation of human PLG. This pathomechanism can be of relevance in both non-invasive and invasive infections where fibrin and other non-fibrin substrates are targeted by increased activity of plasmin. To protect against entrapment by human host fibrin, *S. pyogenes* has evolved to (i) recruit PLG and PLM to its surface, (ii) secrete streptokinase to directly catalyse the production of PLM, and (iii) increase PLM activity by secreting SLO. SLO co-operates with host and bacterial PAs to accelerate the conversion of PLG to PLM. PAs plasminogen activators, PAM plasminogen-binding group A streptococcal M-like protein, SKA streptokinase. **c** created with Biorender.com.

that SLO functionally resembles the co-factor/enhancer role previously shown for PAM and α-Enolase, by synergistically co-operating with SKA or exploiting host PA to generate PLM. Throughout the course of infection, the increased PLM activity can then be exploited by GAS to prevent fibrin entrapment, and also to facilitate tissue invasion, immune evasion and systematic dissemination within the host (Fig. 6a).

Human PLG plays a central role in the coagulation-fibrinolysis system and is a target for several PLG-binding bacterial proteins[23]. These interactions can modify the activity of PLG to promote bacterial survival by breaking down host fibrin to escape entrapment. To date, five streptococcal proteins have been reported to interact with PLG, among which streptokinase (SKA) and plasminogen-binding M-like proteins (PAM) are the most studied[13,18–20,56]. Unlike SKA, SLO is part of the GAS core genome according to previous studies[57]. Furthermore, the high degree of conservation in the primary structure of the PLG-binding motifs in SLO is in contrast to the high variability observed in the a1a2 motif found in plasminogen-binding M-like proteins (PAM)[58]. Consequently, SLO-mediated enhancement in fibrinolysis transcends serotypes and appears to be a more conserved and prevalent pathomechanism. In fact, previous studies have shown that the emergence of invasive GAS infections is connected to upregulated expression of SLO[59].

In this study, we used two orthogonal MS-based methods to identify and validate the binding interfaces. The distance constraints generated by XL-MS were further used to guide the downstream modelling of the protein complexes. Intriguingly, XL-MS also identified cross-links between SLO and tPA, suggesting that SLO retains PLG and tPA to catalyse the generation of PLM. In addition, the intra-links identified by XL-MS provided insights into the protein dynamics of both PLG and SLO during interaction. The over-length cross-links indicate that domain 4 in SLO is subjected to dynamic movement, where a similar domain movement has recently been reported for the SLO-NADase complex[60]. Given that a static crystal structure may not fully represent the dynamic nature of multi-domain protein dynamics in solution phase, the identified cross-links are likely derived from an ensemble of different conformations, including both bound and unbound states. Reciprocally, the clusters of over-length cross-links within PLG specifically pinpoint local movement in the PAN, K4 and PSD domains, which is not necessarily feasible in SAXS due to the presence of SLO as a macromolecular protein binder.

We initially speculated that binding of SLO to PLG would induce a large conformational change in PLG to a relaxed state to facilitate proteolytic processing by PA. However, the HDX-MS data demonstrated that PLG does not undergo major conformational change, at least not within a 9000 s temporal scale of deuteration used in this study. It should be noted that the disulphide bonds within PLG limit the sequence coverage in the HDX-MS analysis precluding a complete analysis of all regions in PLG. Nevertheless, HDX-MS identifies K2 and PSD domains as the primary SLO binding domains that overlap with residues derived from XL-MS computational modelling. In addition, we observe an exosite for SLO-binding similar to what has been reported for the SKA and PAM interaction with PLG[61]. The significant reduction of intra-links found within PLM compared to PLG indicates a major conformational change upon formation of PLM. This change may separate the primary and secondary binding sites (exosite) in space, leading to a loss of SLO-binding as shown by the ELISA and XL-MS results.

The SLO-centric protein network presented here proposes previously unidentified associations with other human plasma proteins like APOE and CLU. APOE contains five O-linked glycosylation sites, while CLU has six N-linked glycosylation sites. Since cholesterol-rich eukaryotic membranes are primary targets for SLO[62,63], it is plausible

that SLO isolates APOE through interactions with cholesterol or glycans. A similar mechanism might apply to CLU, an enigmatic glycoprotein known for its broad range of glycan interactions[64,65], which aligns with SLO's ability to bind various glycans like lacto-N-neotatraose and blood group B type IV pentasaccharide on cell surfaces[35]. Glycoproteomic reanalysis of the AE-MS data revealed significant enrichment of APOE and CLU glycopeptides in pulldown samples when using tagged SLO as bait. Recent reports indicate that human serum albumin interacts with SLO and inhibits the haemolytic effect[38]. In our study, we did not identify albumin as a significant SLO-binder, possibly due to low affinity or from competitive binding by the circulating SLO-specific antibodies. The AE-MS results show significant enrichment of immunoglobulin peptides demonstrating that healthy plasma has abundant circulating IgG clones directed against SLO as previously reported[66], however, to what extent these specific antibodies correlate with protection remains unexplored.

An unsolved question is to what extent SLO contributes to increased plasmin activity in vivo. Further investigations are needed to clarify the nature of these interactions and the implications for GAS pathogenicity. Considering the low efficacy of tPA under normal conditions[58], the enhanced catalytic effect facilitated by SLO may hold translational therapeutic value[67]. For example, the development of mimic peptides based on these defined binding motifs could be explored as a potential approach to manipulate thrombolysis. Notably, peptides derived from the binding interface of a newly identified cellular receptor (mannose receptor MRC-1) for cytolysins have shown effective in vivo protection against bacterial infections[68]. To summarise, the discovery of this indirect PLG activation pathomechanism sheds light on the multifunctional nature of SLO. The moonlighting role proposed here augments the intimate relation between GAS pathogenicity and the host coagulation/fibrinolysis system. Our findings highlight the potential of MS-based proteomics in discovering and clarifying host-pathogen interactions, advancing the understanding of GAS virulence strategies and opening up potential avenues for therapeutic targets against GAS infections.

## Methods

### Ethics statement
In this study, IgG isolate was purified from the plasma of a donor who recently recovered from a GAS infection. This study was conducted in accordance with the principles of the WMA Declaration of Helsinki and complies with all relevant regulations regarding the use of human participants. Written informed consent was obtained from the donor involved and the research was approved by the regional ethics committee (permit number 2015/801).

### Recombinant bait protein production
A composite affinity-tag, comprising hex histidine, streptavidin-binding peptide and hemagglutinin sequences, was genetically engineered onto the N-terminus of SLO/GFP and the C-terminus of SCPA (Supplementary Notes). The tagged GFP and tagged SCPA proteins were synthesised, produced, and purified at the Lund University Protein Production Platform (LP3). Tagged SLO proteins were recombinantly expressed and purified in-house. The plasmids encoding the tagged SLO fusion were synthesised and assembled by LP3, then subsequently introduced into BL21(DE3) Competent Cells (Thermo Fisher Scientific). Following o/n induction with 0.01 mM IPTG (Isopropyl ß-D-1-thiogalactopyranoside; Sigma–Aldrich) and protein extraction with BugBuster (Novagen), the target tagged SLO protein underwent $Ni^{2+}$-charged IMAC resin (Bio-Rad) purification, following an established protocol[24] alongside the manufacturer standard procedures. Verification of the bait proteins' purity and sequence integrity was conducted by SDS-PAGE and bottom-up mass spectrometry (BU-MS), ensuring the successful production of the bait proteins.

## Affinity-enrichment mass spectrometry (AE-MS) and data analysis

Pooled human plasma, sourced from the Innovative Research company, was diluted with phosphate-buffered saline (PBS tablet; Sigma−Aldrich). The affinity-enrichment coupled with mass spectrometry (AE-MS) was performed in line with the workflow previously described in literature[41]. In summary, 40 μg of the tagged bait proteins were immobilised onto a 150 μL 50% slurry Strep-Tactin Sepharose resin (IBA Lifesciences GmbH), within a Bio-Rad spin column for each experiment. The immobilised proteins were incubated with either undiluted or 1x PBS-diluted human plasma (50%, 10% HP) at 37 °C and 500 rpm for 2 h, followed by rigorous washing with eight column volumes of ice-cold 1x PBS buffer. Bait-enriched prey proteins from plasma were then eluted with 5 mM biotin (Sigma−Aldrich) in ice-cold 1x PBS. The affinity-enrichment experiment was independently repeated three times for each bait protein at various dilutions of the prey mixture. Biotin removal was achieved through protein precipitation by adding trichloroacetic acid (TCA; Sigma−Aldrich) to a final concentration of 25%, and samples were incubated at −20 °C overnight. After centrifugation at $18,000 \times g$ for 30 min at 4 °C, the supernatant was discarded, leaving the pellet, which was subsequently washed with cold acetone (Thermo Fisher Scientific). The resulting pellets were resuspended in 50 μL of 8 M urea with 100 mM ammonium bicarbonate (Sigma−Aldrich), then subjected to reduction with 5 mM TCEP (tris(2-carboxyethyl)phosphine; Thermo Fisher Scientific) and alkylation with 10 mM IAA (iodoacetamide; Sigma−Aldrich), followed by an overnight digestion at 37 °C and 500 rpm in a ThermoMixer (Eppendorf) using trypsin (Promega) at a 1:20 enzyme-to-substrate ratio. The digested peptides were cleaned up using a C18 spin column (Thermo Fisher Scientific), concentrated and dried via SpeedVac (Eppendorf), and reconstituted in buffer A (2% acetonitrile, 0.2% formic acid; Thermo Fisher Scientific) for mass spectrometry analysis.

For the liquid chromatography-mass spectrometry (LC-MS) analysis, ~800 ng of peptides from each experiment, as determined by a NanoDrop spectrophotometer (DeNovix), were loaded onto an EASY-nLC 1200 system interfaced with a Q Exactive HF-X hybrid quadrupole-Orbitrap mass spectrometer (Thermo Fisher Scientific). Each sample was injected twice, once for data-dependent acquisition (DDA) and once for data-independent acquisition (DIA). Initially, peptides were concentrated using a PepMap100 C18 pre-column (75 μm × 2 cm, with 3 μm particle size; Thermo Fisher Scientific) and subsequently resolved on an EASY-Spray column (ES903; Thermo Fisher Scientific) maintained at 45 °C, following the manufacturer's instructions. The chromatographic separation was executed using two mobile phases: Solvent A with 0.1% formic acid and Solvent B comprising 0.1% formic acid in 80% acetonitrile. A linear gradient was applied, ranging from 3% to 38% of Solvent B across a 120-min period, with a flow rate held steady at 350 nl/min. For DDA strategy, it initiated with an MS1 scan covering a m/z range of 350–1650, at a resolution of 120,000 and an AGC target of 3e6, with a maximum injection time (IT) of 45 ms. This was succeeded by the top 15 MS2 scans, each with a resolution of 15,000, an AGC target of 1e5, a 30 ms IT and a normalised collision energy (NCE) of 28. Charge states of 1+, 6 to 8+, and higher were excluded. For DIA approach, it mirrored the DDA LC settings but in MS analysis adopted an MS1 scan over a m/z range of 390–1210, at a resolution of 60,000, an AGC target of 3e6 and a maximum injection time of 100 ms, followed by MS2 scans within a fixed isolation window of 26.0 m/z. Here, a resolution of 30,000, an AGC target of 1e6 and an IT of 120 ms were applied, along with a NCE of 30. To assure system performance, yeast protein extract digest (Promega) standards was analysed throughout the measurement.

The initial batch of twenty-seven DDA datasets were searched against a human reference proteome database including common contaminants (UPID: UP000005640) using MaxQuant software (version 2.0.3.0) with the default settings[69] to construct a spectral library.

Based on this, nine DIA datasets were processed using the MaxDIA workflow[70]. pGlyco 3.0[71] was used to identify glycopeptides of PLG from AE-MS DDA runs. Label-free DIA quantification raw intensity were processed through the MiST score pipeline[44,72], where bait-prey interaction with a MiST score > 0.5 were selected for the initial screening. On the other hand, normalised MaxLFQ intensities from both DIA and DDA were processed in Perseus[73] to remove any invalid data points (common contaminants, decoy identification, etc.), $\log_2$-transform, and perform differential statistical analysis between bait groups. MiST scoring metrics, FDR-corrected multiple t-tests (two-side) coupled with fold change threshold, and two-way ANOVA multiple comparisons (Tukey's post-hoc test) were used to cross-validate significant plasma protein binders to specified baits. Row-level (among sample replicate) z-score normalisation is applied for each protein group across different samples. 68 MiST-derived SLO-interactive proteins were queried for identifying prey-prey link using built-in STRING[46] High-Confidence Human Protein Links (score ≥ 0.7) interactome database function via Cytoscape[74]. For other comprehensive data visualisation, tools such as Cytoscape[75], and R packages including ggplot2 and pheatmap were employed. Furthermore, the Metascape tool[76] was utilised for performing over-representation (enrichment) analysis to explore the association among isolated proteins and to interpret the biological significance of the data.

## Indirect ELISA assessment

For the quantification of specific IgG titres via ELISA, same amounts of SLO protein or SCPA were coated to MaxiSorp plates (Thermo Fisher Scientific) and incubated overnight at 4 °C. Non-specific binding sites on the plates were then blocked with 1% bovine serum albumin (BSA; Sigma−Aldrich). Subsequently, the plates were added with a two-fold serial dilution of 10 μg of either IVIG (pooled IgG isolates from healthy population), P.IgG (IgG isolates from the plasma of a donor who recently recovered from a GAS infection as described before), Xolair (an anti-IgE IgG; Novartis), or 1x PBS as a background reference. After a thorough washing, 100 μL of a 1:3000 dilution of HRP-conjugated protein G (Bio-Rad) and 100 μL of HRP substrate buffer (20 ml NaCi-trate pH 4.5 + 1 ml ABTS + 0.4 ml $H_2O_2$; Sigma−Aldrich) were sequentially mixed with samples. The reaction developed over 30 min and absorbance was read at 450 nm. For indirect ELISA aimed at validating the purified native PLG/PLM (Thermo Fisher Scientific) binding specificity for SLO, equivalent quantities of PLG or PLM were first immobilised and then incubated with SLO protein, followed by a diluted mouse monoclonal anti-SLO antibody (Abcam) and an anti-mouse HRP-conjugated secondary antibody (Bio-Rad). The procedures for substrate development and absorbance reading were the same as detailed above.

## Plasminogen activation and plasmin activity assay

The assays to evaluate plasminogen activation and plasmin activity were conducted using a colorimetric quantification kit from Abcam. The PLG activation assay employed 96-well flat bottom microplates and a mimic chromogenic substrate, whose hydrolysis by plasmin at 25 °C led to the release of p-nitroaniline. This distinctive final product was then quantified by measuring absorbance at 405 nm by a BMG Labtech microplate reader. To assess PLG activation rate, SLO was incrementally added to the PLG samples across specified concentrations in four experimental replicates. Following the addition of SLO, the mixture were incubated for 30 min in a ThermoMixer to facilitate interaction. Subsequently, plasminogen activators (either tPA, Abcam or SKA, fisher scientific) were introduced along with the mimic plasmin substrate. Data obtained from the assay was analysed with GraphPad Prism software (version 9.0), where initial activation velocities were derived from the linear part of the absorbance versus time-squared plot ($mA_{405nm}/t^2$). For the plasmin activity assay, an inhibitory mix was included to negate any non-specific background that might be present

in the samples. The influence of SLO on plasmin activity was examined by the incremental addition of SLO to the reaction, following the same processes as the PLG activation assay for the substrate addition and subsequent absorbance measurement.

### SLO cytolysis inhibition assay

In assessing the inhibitory effect of PLG on SLO-mediated cytolysis, sheep red blood cells (Thermo Fisher Scientific), were initially diluted with 1x PBS to 5% v/v suspension. To this diluted RBC suspension, 100 ng of TCEP-reduced active SLO protein and 5 µg of PLG/PLM proteins were added. Xolair and P.IgG were also included in the assay as negative and positive control separately. The mixtures were incubated in a ThermoMixer at 37 °C 300 rpm for 30 min. Then, the reaction microplates were centrifuged, and the supernatant was transferred to fresh plates for absorbance measurement at 541 nm wavelength using a BMG Labtech microplate reader, which corresponds to the free haemoglobin concentration indicative of cell lysis. A 100% lysis rate was set using the $A_{541nm}$ from the group of 0.6% RIPA (Thermo Fisher Scientific) lysis buffer.

### Hydrogen/deuterium exchange mass spectrometry (HDX-MS) and data analysis

The Hydrogen/Deuterium Exchange Mass Spectrometry (HDX-MS) setup involved the use of a LEAP H/D-X PAL™ system for automated sample preparation, connected to an LC-MS system comprising an Ultimate 3000 micro-LC linked to an Orbitrap Q Exactive Plus MS. HDX experiments were conducted on plasminogen (PLG) in both its unbound (apo) and SLO-bound (complex) states, using a 1:2 molar ratio mixture of PLG to SLO in 10 mM PBS. The samples were labelled for various intervals (0, 30, 3000 and 9000 s) at 4 °C in PBS or a $D_2O$-based HDX labelling buffer (Thermo Fisher Scientific). Each state from each labelling time interval repeated three times within a single continuous run. The labelling reaction was quenched by diluting the samples with 1% trifluoroacetic acid (TFA; Thermo Fisher Scientific), 0.4 M TCEP and 4 M urea, all maintained at 4 °C. Following quenching, the samples were injected at 4 °C, in a flow rate of 50 µL/min 0.1% formic acid for 4 min on-line digestion and trapping of the samples. The digested peptides were then subjected to a solid-phase extraction and washing process using 0.1% formic acid on a PepMap300 C18 trap column (Thermo Fisher Scientific), subsequently switched in-line with a reversed-phase analytical column (Hypersil GOLD). Chromatographic separation was performed at 1 °C, employing a mobile phase gradient from 5 to 50% of Solvent B (95% acetonitrile/0.1% formic acid) over 8 min, and then from 50% to 90% over the next 5 min. The separated peptides were analysed on the Q Exactive Plus MS, which featured a heated electrospray ionisation (HESI) source operating at a capillary temperature of 250 °C. Full scan spectra were obtained with a high resolution of 70,000, using an automatic gain control target of 3e6 and a maximum ion injection time of 200 ms, covering a scan range of 300–2000 m/z. Peptide identification was achieved by analysing digested un-deuterated control PLG peptide samples through data-dependent acquisition tandem MS/MS. A comprehensive peptide library, including peptide sequence, charge state and retention time information, was prepared for HDX analysis using PEAKS Studio X (Bioinformatics Solutions Inc.), searching pepsin-digested, un-deuterated samples against the PLG sequence (UniProtID: P00747). HDX-MS data were then processed and analysed using HDExaminer v3.1.1 (Sierra Analytics Inc.). HDX data summary is reported in Supplementary Table 4.

Comparative analysis of PLG in SLO-bound state versus apo state was conducted using individual charge states for each identified peptide. Deuterium incorporation levels were determined based on the observed mass differences between deuterated and undeuterated control peptides, without back-exchange correction using fully deuterated samples for comparison. Manual inspection of the spectra was

carried out to eliminate low-scoring peptides, obvious outliers and those with inconsistent retention times. Deuteros 2.0 software[77] was applied to perform both FDR-controlled significance tests and hybrid significance tests and to demonstrate changes in deuterium uptake through kinetic uptake, barcode, redundancy, coverage, butterfly, woods and volcano plots. This software also contributed to the projection of the coordinates of protected peptide residues onto the reference crystal structure of PLG (PDB: 4DUR), aiding in the interpretation and visualisation of binding interactions and protein dynamics.

### Cross-linking mass spectrometry (XL-MS) and data analysis

For the cross-linking experiment, in-solution cross-linking was conducted on PLG and SLO, PLM and SLO, as well as PLG and tPA combinations, each at a 1:1 molar ratio, and for a three-protein mixture of PLG, SLO and tPA at ratios of 1:1:1 and 1:2:1. These proteins were mixed in 1x PBS and incubated at 37 °C in a ThermoMixer with agitation at 500 rpm for 1 h. Cross-linking agents, specifically DSS-H12/D12 and DSG-H6/D6 from Creative Molecules, were then individually introduced to cross-link the protein mixtures for 2 h. The reactions were quenched with 4 M ammonium bicarbonate (Sigma–Aldrich), followed by conventional reduction and alkylation steps as detailed before. A two-step protease digestion scheme including (i) 2-h lysyl endopeptidase (FUJIFILM Wako Chemicals U.S.A. Corporation) digestion and (ii) o/n trypsin (Promega) digestion was then performed to generate the cross-linked peptides.

Once prepared, the peptides were subjected to a clean-up process, dried and resuspended prior to MS analysis. An Orbitrap Eclipse Tribrid Mass Spectrometer (Thermo Fisher Scientific), linked to a Nanospray source and combined with an Ultimate 3000 UPLC system (Thermo Fisher Scientific), was used for LC-MS detection for cross-linked peptides. Approximately 600 ng of peptides, quantified using a NanoDrop spectrophotometer, were loaded onto a PepMap RSLC column (Thermo Fisher Scientific) maintained at 45 °C for concentration. Duplicate injections were carried out for each sample. Column preparation and peptide loading adhered strictly to the manufacturer's guidelines. The chromatographic separation utilised a gradient of Solvent A (0.1% formic acid) and Solvent B (0.1% formic acid in 80% acetonitrile), progressing linearly from 4 to 38% Solvent B over 90 min at a flow rate of 300 nl/min. The mass spectrometer operated in positive DDA mode, with an MS1 scan ranging from 400–1600 m/z at resolution of 120,000; a standard-mode AGC target and auto-mode maximum injection time, followed by a series of MS2 scans following a set cycle time of 3 s; 15,000 resolution; standard AGC target; 22 ms IT and NCE of 30. Charge states ranging from 2 to 6+ were included in the analysis. LC-MS system performance was benchmarked using HeLa protein digest standards (Thermo Fisher Scientific) throughout the process.

The databases for searching cross-linked peptide encompassed PLG/PLM (UniProtID: P00747), tPA (UniProtID: P00750) and SLO sequences (UniProtID: P0DF96), in addition to common contaminants. The search parameters regarding modification related to cross-linking were configured as specified by the manufacturers. Fixed modifications were assigned to carbamidomethylated cysteines while acetyl (protein N-term) and oxidation (M) was marked as a variable modification. Data analysis was conducted using pLink 2 (version 2.3.11) and MaxLynx (version 2.0.3.0) software to extract high-confidence spectrum evidence, achieved by reducing the maximum allowed missed cleavages of peptides from 3 (default) to 2 with the rest of parameters kept as previously recommended[78,79]. The selected cross-linked peptide MS/MS spectra was then visualised using MaxLynx. DisVis analysis[80] was performed to visualise the interaction space of tPA using all tPA-PLG-SLO heterotrimeric protein complex conformation consistent with the identified interprotein cross-links between tPA and SLO, with Cα-Cα distance set to be within 0-40 Å for comprehensive profiling. XL-MS data summary is detailed in Supplementary Table 5.

## Computational modelling of pairwise protein complex and data analysis

The computational modelling was conducted following previously established methods[81]. In summary, the AlphaFold2-Multimer[52] was utilised for the initial prediction of the PLG PSD-SLO D4 complex structure without incorporating experimental data. The NMSim[53] analysis was applied to assess the plausible motions of the SLO domains resulting in 5 trajectories of 500 conformers. For the prediction of SLO conformational changes, these 2500 conformers were investigated and the best that matched the intraprotein cross-links were selected. As for modelling PLG-SLO pairwise complex, the Rosetta[51] software was used to generate two sets of 10,000 models based on different SLO conformational states (native form, and bent form) through the Rosetta docking protocol[51], with the highest-scoring models being chosen for subsequent analysis. The MS/MS analysis workflow from TX-MS[40] technique was then employed to select top-ranking models based on distance constraints derived from XL-MS datasets. Finally, a high-resolution refinement of the top-ranking models was performed using the RosettaDock protocol[82] to repack sidechain. Residues in the binding interface of selected models were determined by PRODIGY[83], highlighted with sidechain in stick presentation and zoomed in with covalent bonds formed by linker connecting reactive lysine side chains in proximity. ChimeraX[84] was used to generate the final figures.

### Reporting summary

Further information on research design is available in the Nature Portfolio Reporting Summary linked to this article.

## Data availability

All mass spectrometry proteomics data have been deposited to the ProteomeXchange Consortium via the PRIDE[85] partner repository with the dataset identifier PXD051261. Source data are provided with this paper.

## Code availability

Scripts used for the data analysis and result visualisation of this paper are available at Zendo repository under accession code 1321267.

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

## Acknowledgements

We gratefully acknowledge the SciLifeLab Integrated Structural Biology (ISB) Platform, the Swedish National Infrastructure for Biological Mass Spectrometry (BioMS) and the Lund Protein Production Platform (LP3) for providing facilities and experimental support. We would also like to thank Dr. Hong Yan, Dr. Tommaso De Marchi, Dr. Alejandro Gomez Toledo, Dr. Annika Rogstam and Dr. Wolfgang Knecht for assistance. J.M. is a Wallenberg academy fellow (KAW 2017.0271) and is also funded by the Swedish Research Council (Vetenskapsrådet, VR) (2019-01646 and 2018-05795), the Wallenberg foundation (KAW 2016.0023, KAW 2019.0353 and KAW 2020.0299), and Alfred Österlunds Foundation. H.K. was supported by the French Agence Nationale de la Recherche (ANR), under grant ANR-22-CPJ2-0075-01.

## Author contributions

D.T., L.H. and J.M. conceptualised the manuscript. The methodology was jointly designed by D.T., H.K., S.E., L.H. and J.M. D.T. conducted the laboratory experiments and prepared samples for mass spectrometry analysis. The data analysis and interpretation were carried out by D.T., H.K., L.M., S.E., L.H. and J.M. Computational modelling was designed and conducted by H.K., and bioinformatic analyses were performed by E.H. and L.M. D.T. made the figures. J.M. supervised the work and secured the funding. D.T. and J.M. wrote the initial draft of the manuscript, with critical input from H.K., E.H., L.M., S.E. and L.H. All authors have reviewed and approved the final version of the manuscript.

## Funding

## Competing interests

The authors declare no competing interests.
