## [Transparent Peer Review file · Nature Communications]

Streptolysin O Accelerates the Conversion of Plasminogen to Plasmin

Corresponding Author: Mr Di Tang

Version 0:

Reviewer comments:

Reviewer #1

(Remarks to the Author)

The manuscript by Tang et al describes a new function for the *S. pyogenes* haemolysin streptolysin O (SLO) in promoting activation of the host protease plasminogen (PLG) by tissue plasminogen activator (tPA). The authors first used affinity enrichment and mass-spectrometry to identify PLG as an SLO binding partner. They then used a variety of mass spectrometry and biochemistry approaches to characterise this interaction, demonstrating that SLO increases the rate of PLG activation by tPA. Their data supports a model that SLO induces localised conformational changes in PLG that increases accessibility of the tPA cleavage site which promotes conversion of PLG to the active protease plasmin.

The authors seem to have found a novel mechanism for activation of PLG by *S. pyogenes*, which is an important virulence property of this pathogen. I find the work very interesting, but I must emphasise that my expertise does not extend to the mass spec approaches utilised in this study. However, from the perspective of my own expertise in *S. pyogenes* biology I find the work intriguing and novel. The suggestions below are primarily to improve the readability of the manuscript for a wider audience and to contextualise findings with what is known about *S. pyogenes* biology.

Major points:

A. Readability for a broader audience

1. The authors have done a nice job describing their findings, but as a reader outside of the mass spectrometry field I found the manuscript very technical in sections. Readability for broad audience of this journal could be improved for some sections, particularly the HDX-MS section. Figures are also quite technical and don't always contain clear descriptions in the figure legends. I have detailed several instances of these below under the "Figures" heading.
2. I also suggest including some basic descriptors and/or diagrams for the different MS methods to extend readability for audience not familiar with these methods (similar to the diagrams included for indirect ELISA in Fig 1C, and the PLG activation assay in Fig 1E, both are quite simple techniques by comparison). These could be included in the supplementary.

B. Contextualising results with the expanding roles of SLO in GAS biology

3. The authors have framed their findings around roles of plasminogen activation in invasive disease (e.g. as depicted in Fig 6C). However, SLO is especially important for superficial disease, and increased expression has been associated with the emergence of successful pharyngitis clones (e.g. M1T1, M3, M89 - see PMID 30109840). As both PLG and tPA are also abundant in the throat, I suggest including mention of this and potentially PLG activation and degradation of other factors in addition to fibrin clots at the mucosal surface (e.g. ECM).
4. It is interesting why *S. pyogenes* would have an alternative way to activate plasminogen independently of SKA, when this method is conserved, highly effective and tightly controlled. The authors don't raise this until the final paragraph, but it should be fairly easy to test along the lines of how they have done in Figs 2F+G. If SLO similarly increases PLG activation by SKA, then this supports a model where SLO is a general promoter of PLG activation. If it does not, or is detrimental, this suggests that *S. pyogenes* coordinates PLG by unique mechanisms in different niches. Both scenarios would be interesting to the wider *S. pyogenes* and bacterial pathogenesis fields.
5. I am also curious whether PLM activation by SLO and tPA leads to inactivation of SLO. The authors show nicely that haemolytic activity was not affected by PLG, but this was done in the absence of tPA. Does addition of tPA affect haemolysis?

C. Data and Code availability

6. Please include source data files for all data presented in graphs within the figures. Given that the data generated in this study will be very useful to the larger community, in addition to submitting the raw data to ProteomeXchange consortium, the authors should consider making their processed data for proteomics analysis (for instance, normalised log2 transformed tables as CSV table) and data used to generate the figures easily available. These non-restricted data can be provided as supplementary material or through third party hosting options. Authors should provide a spreadsheet as supplemental data for differential expression comparisons to accompany proteomics data represented in the figures as it is hard to interpret from plot visualisation alone.

7. In the interests of data reproducibility, transparency and open science, the authors should also share their scripts for performing the computational analysis related to proteomics experiments shown in this manuscript in a code sharing platform like GitHub/GitLab etc.

D. Minor comments

line 66: surface dehydrogenase is the same protein as GAPDH

lines 81-83: Suggest including reference to roles of SLO in translocating NADase into cells (Madden et al. PMID 11163247) and in activating superantigen SSA (Brouwer et al. PMID 33024089).

Line 108: Add a phrase justifying why GFP was chosen as a reference control.

Line 118: List relevant IgG chains that were specifically enriched by SLO or SCPA.

Line 142: The statement that only PLG was significantly enriched in 10% plasma seems to contradict slightly the figures. While PLG is the most significant of the three proteins (of PLG, APOE and CLU), the fold change is lower than for APOE and similar for CLU, and all three proteins are above the log P value threshold of -2.

Line 328: This sentence requires a reference.

Lines 351-2: My understanding is that SKA is encoded by all GAS strains, as it is part of the core genome (see: <https://www.biorxiv.org/content/10.1101/2023.08.29.555273v1.full>).

Lines 352-354: References should be cited to support these statements.

E. Methods

Lines 413-416: Please provide details about the protein constructs, including primary sequence and the exact residues included for SLO and SCPA. Both proteins possess an N-terminal signal peptide and SCPA also has a C-terminal sorting signal that is cleaved to form the mature polypeptide.

Line 476: More details should be provided on parameters used for STRING network. For example, version used, rationale of proteins selected for network (i.e. proteins with differential abundance?), how were the proteins grouped, what active prediction methods were used, interaction scores, etc.

Line 476-477: Add software versions for Cytoscape, STRING (website), R and Metascape.

Line 487: Include ethics details of plasma from donor who recently recovered from GAS infection.

F. Figures

Fig 1A: reference domain structure for PLG in the figure legend. Or preferably, include a more detailed figure in the Supplementary showing the domains and amino acid numbering that defines these, and the tPA cleavage site. Also, black font on dark purple circles is hard to see.

Fig 1B: Green circles are not defined in the Key. Also this figure appears to be very similar to Extended Data Fig 3. Are these the same? If so, this should be referenced in the Figure legend. Also it is difficult to distinguish between the colours representing 'regulation of complement activation' and 'reverse cholesterol transport.' Consider formatting to make it clear.

Fig 2A: Is this the 10% diluted plasma? This should be specifically stated in the figure legend.

Fig 2D: I am confused why the pre-coating with SLO gives such a low ELISA value with anti-SLO mAb (as shown in Fig 2C). Wouldn't the primary antibody bind the SLO used for coating? Or is this figure mislabeled?

Fig 2F: the different SLO conditions (quantity or time?) are not annotated. Presumably these are the same as in Fig 2G but should be annotated in this figure also.

Extended Data Fig 5A - Include tPA cleavage site

Fig 3A: While I am not familiar with these plots, the APO and SLO-bound plots look very similar to me (as noted by the authors). I assume the differences reflect the magnitude of the DU values in the boxed regions? Also, the amino acid ranges in the text on line 187 don't match the annotations in the figure (e.g. "regions spanning residues 74-84 in PAN" is annotated as "74-88" in Fig 3A). Are these referring to the same thing? Same goes for Lines 192-3 and Extended Data Fig 5B.

Extended Fig 1: The authors should comment on the fact that one GFP bait control is different. Is there a reason why this control is different from the others?

Extended Fig 2A: Black font on dark purple circles is hard to see. Also, according to ref. 36, a MiST score is used to identify the biologically relevant interactions. The referenced considered interactions with a MiST score greater than 0.75 as significant. Please include the value that was used for your analysis.

Extended Data 5D: include tPA cleavage site

Fig 5F: Numbering in figure (192-203) doesn't match what is stated in the text (194-203) on line 312.

Reviewer #2

(Remarks to the Author)

Reviewer #3

(Remarks to the Author)

In this manuscript, Tang et al use an integrative structural proteomics strategy to interrogate the interaction of SLO with PLG. My expertise is in the field of structural proteomics and my comments below reflect this. Overall, I found some of the data presentation confusing and it was difficult for the reader to understand how the data supported the conclusions made – I am sure this comes down to presentation and the authors should improve this upon revision.

- Line 105 – indicate the affinity tag used.
- The authors use statistical analyses to identify significant SLO interactions (line 128). How was significance determined? Did a protein have to pass their thresholds in both analyses SLO/GFP and SLO/SCPA to be significant?
- Line 144. The authors state their AE-MS identified three binding partners. Whilst their analysis focused on three, im not sure that ONLY three were detected. Could the authors please clarify?
- In Extended Data Figure 2, Figure 1 A, the colours of the nodes should be better explain.
- What is a MiST score? And why do the width/colour of the edges reflect this? Some explanation is needed in the text/figure legends.
- In Extended Data Figure 1, what statistical test was performed here for proteins to be shown in this heatmap? More information in the legends is needed.
- The authors choose to characterise the interaction of plasminogen with SLO by HDX-MS and XL-MS. Is the affinity of the interaction known? In the HDX experiments what % bound plasminogen was used to prepare the samples. This is vital knowledge to correctly design these experiments.
- Plot in Figure 3A. The authors state in the text (line 187) that minimal differences between states were observed except in three key regions. It is unclear to me how they came to such a conclusion from the data presented. Was a statistical analysis performed? It appears to me that there are other peptides with differences in uptake of comparable levels to these three regions. Why are these three regions indicated? A presentation of these data as a Wood's plot would be easier to interpret, and the statistical analysis could be presented here too. This analysis also seems inconsistent with the data they present in Figure 3B. Why did the authors choose to plot the data for only the PAN and PSD domains in Figure 3E, given that they see other changes in other regions of the protein. It would be beneficial to plot some of these changes onto structure etc of the proteins.
- Why do the authors study PLG-SLO and PLM-SLO by XL-MS, but only PLG-SLO by HDX-MS?
- Of the identifications with the different software packages used for XL-MS, how many were unique, and why did the authors only choose to visualise the pLink results? (line 226)
- Only one XL was detected and the authors propose that this is likely because the interaction is weak. How do the authors define weak, without a Kd measurement? Are there Lys residues in/adjacent to protected regions from HDX, that might be crosslinkable, or is this experimental design flawed by the choice of the wrong crosslinking reagents? (or proximal Lys residues in the structural models shown later)
- The authors should make it clear what concentrations of crosslinker was used, and show SDS-PAGE gels demonstrating the quantity of crosslinked material and the quality of their experimental design. This should be the case for all crosslinking reactions.
- The authors state that the 'fraction of overlength Ca-Ca distances were higher in SLO c.f. PLG (line 252, Figure 4C). From the figure it is hard to justify this statement. It might be more correct that more overlength crosslinks exceed ca 75 Å? Is this what the authors are referring to? If so, how many crosslinks fit this criterion?
- The authors then interpret these results in the context of SLO dynamics. It is not clear to me how they draw the conclusion made (line 255), without comparing the same data between the apo and bound states.
- What do the authors mean by targeted crosslinking (line 275)? Is this referring to the computational procedure used? If so, this is misleading. This whole latter section of the results and the way the data are presented in Figure 5 were very difficult to digest and understand. The authors should work to improve the clarity of this complex analysis.
- The authors state that the AlphaFold model could not predict the correct orientation of SLO with respect to PLG or explain the inter-links detected... (line 284). Were the predictions of these regions poor (i.e. pLDDT scores low), or did the model not agree with their data? The authors should show data that would support both of these situations and explain which is true.
- Line 297. How many structures were consistent with most crosslinks? Was this a rare event? How many crosslinks were not satisfied? Were these the 4 crosslinks that were consistent with the alternate conformer proposed in Figure 5E?
- Figure 5F – not clear how this relates to the statement on line 311. Surely it would be better to do a side-by-side comparison of HDX data with the predicted interface.
- Community standards for reporting HDX [Nature Methods, 16, 595–602 (2019)] and XL [Anal. Chem. 91, 6953–6961 (2019)] data should be followed.

Version 1:

Reviewer comments:

Reviewer #1

(Remarks to the Author)

The authors have done a fantastic job in clarifying my questions and conducting additional experiments to address my comments. The manuscript has greatly improved.

I only have one minor comments left to edit the reference list for consistency. For example, inclusion of DOIs; Sentence vs title case (e.g. Ref 27) vs uppercase (Ref 58); symbols at the end of the manuscript title (e.g. Refs 14, 17, 19, 26 etc).

Reviewer #2

(Remarks to the Author)

Reviewer #3

(Remarks to the Author)

The authors have addressed all of my comments in this revision.

Point-by-point Response to Reviewer comments

Dear editor and reviewers,

We sincerely thank the reviewers for their constructive feedback. We have addressed all the comments, performed the suggested experiments, and revised several figures and parts of the text according to the reviewers' suggestions. We believe that the manuscript has been substantially improved after addressing the reviewers' comments.

The major changes include the addition of new data that demonstrate that streptolysin O (SLO) can also enhance the activity of streptokinase (SKA). This finding is notable as it suggests that SLO can form a multiprotein host-pathogen complex together with plasminogen (PLG) and SKA. In addition, we have improved the readability of our manuscript for a broader audience, focusing particularly on the hydrogen/deuterium exchange mass spectrometry (HDX-MS) section and the data presentation parts. We have also improved the figure legends, added a schematic overview of the main MS-based methods and have appended two community-standard HDX-MS and XL-MS data summaries in the Supplementary Information file. Modified parts have been highlighted in blue in the revised manuscript.

Please see below for the detailed point-by-point responses to all reviewer comments.

Kind regards,

Di Tang and Johan Malmström on behalf of all authors

Reviewer #1 (Remarks to the Author):

The manuscript by Tang et al describes a new function for the *S. pyogenes* haemolysin streptolysin O (SLO) in promoting activation of the host protease plasminogen (PLG) by tissue plasminogen activator (tPA). The authors first used affinity enrichment and mass-spectrometry to identify PLG as an SLO binding partner. They then used a variety of mass spectrometry and biochemistry approaches to characterise this interaction, demonstrating that SLO increases the rate of PLG activation by tPA. Their data supports a model that SLO induces localised conformational changes in PLG that increases accessibility of the tPA cleavage site which promotes conversion of PLG to the active protease plasmin.

The authors seem to have found a novel mechanism for activation of PLG by *S. pyogenes*, which is an important virulence property of this pathogen. I find the work very interesting, but I must emphasise that my expertise does not extend to the mass spec approaches utilised in this study. However, from the perspective of my own expertise in *S. pyogenes* biology I find the work intriguing and novel. The suggestions below are primarily to improve the readability of the manuscript for a wider audience and to contextualise findings with what is known about *S. pyogenes* biology.

Major points:

A. Readability for a broader audience

1. The authors have done a nice job describing their findings, but as a reader outside of the mass spectrometry field I found the manuscript very technical in sections. Readability for broad audience of this journal could be improved for some sections, particularly the HDX-MS section. Figures are also quite technical and don't always contain clear descriptions in the figure legends. I have detailed several instances of these below under the "Figures" heading.

We have improved the readability of our manuscript for a broader audience with a particular focus on refining HDX-MS section and data presentation. We have improved the clarity of the figure legends to make the manuscript more accessible to non-specialists. Furthermore, we have added a schematic overview of the MS-based methods used herein and added more technical/analytical detail to the "Methods" section to better explain the experimental principles, rationales, and interpretations. See the detailed responses below for more information.

2. I also suggest including some basic descriptors and/or diagrams for the different MS methods to extend readability for audience not familiar with these methods (similar to the diagrams included for indirect ELISA in Fig 1C, and the PLG activation assay in Fig 1E, both are quite simple techniques by comparison). These could be included in the supplementary.

We have added a diagram to **Supplementary Data Figure 1** (see below) to illustrate the integrated workflow of the MS-based proteomics approaches used in our study. The schematic illustration of the indirect ELISA that we displayed in the first version of the manuscript has been moved to the **Supplement Data Figure 6** due to space constraints.

Supplementary Data Figure 1: Tang et al

Supplementary Data Figure 1

B. Contextualising results with the expanding roles of SLO is GAS biology

3. The authors have framed their findings around roles of plasminogen activation in invasive disease (e.g. as depicted in Fig 6C). However, SLO is especially important for superficial disease, and increased expression has been associated with the emergence of successful pharyngitis clones (e.g. M1T1, M3, M89 - see PMID 30109840). As both PLG and tPA are also abundant in the throat, I suggest including mention of this and potentially PLG activation and degradation of other factors in addition to fibrin clots at the mucosal surface (e.g. ECM).

This is a good comment that we overlooked. We have modified **Figure 6c** (as shown below) to illustrate SLO-mediated increased plasmin activity on other non-fibrin substrates, such as extracellular matrix components. In addition, we have modified the discussion section (page 14) and included a reference to PMID:30109840, elaborating on the role of PLG activation and its implications for various stages of infection.

Figure 6c

4. It is interesting why *S. pyogenes* would have an alternative way to activate plasminogen independently of SKA, when this method is conserved, highly effective and tightly controlled. The authors don't raise this until the final paragraph, but it should be fairly easy to test along the lines of how they have done in Figs 2F+G. If SLO similarly increases PLG activation by SKA, then this supports a model where SLO is a general promoter of PLG activation. If it does not, or is detrimental, this suggests that *S. pyogenes* coordinates PLG by unique mechanisms in different niches. Both scenarios would be interesting to the wider *S. pyogenes* and bacterial pathogenesis fields.

This is a very interesting idea, and we would like to thank the reviewer for suggesting this. We have conducted the same experiments using SKA as the PLG activator. Interestingly, SLO does also, in a dose-dependent manner, significantly enhance SKA-mediated conversion of PLG to PLM, as shown in **Figure 2e** below.

Figure 2e

To determine the possibility of simultaneous binding of SKA and SLO to PLG, we superimposed our top-ranked PLG-SLO model onto the existing co-crystal structure (PDB: 1BML) of the PLG_PSD-SKA¹. This analysis revealed that SLO and SKA have different binding sites on PLG and indicates that PLG-SLO-SKA can form a multi-protein host-pathogen protein complex. We have made a new **Supplementary Data Figure 12e** that illustrates this.

Supplementary Data Figure 12e.

Based on these results, we conclude that SLO can synergistically co-operate with both tPA and SKA to exploit plasminogen activators to increase the plasmin activity. We have incorporated these findings into the main text, updating **Figure 2** and the corresponding results section on page 8. Additionally, we revised the abstract and the discussion on page 14, and introduced a new figure in the Supplementary Information file.

5. I am also curious whether PLM activation by SLO and tPA leads to inactivation of SLO. The authors show nicely that haemolytic activity was not affected by PLG, but this was done in the absence of tPA. Does addition of tPA affect haemolysis?

Our cross-linking data indicate a very weak interaction between tPA and SLO, supported by fewer cross-link evidence in both evidence count and intensity. Specifically, there are only 7 cross-link spectrum matches for tPA-SLO compared to 41 for PLG-SLO in the corresponding PLG-SLO-tPA cross-linked datasets. Consequently, these results suggest that tPA binding to SLO is not strong enough to inhibit the haemolytic activity of SLO. We hypothesize that cross-links between SLO and tPA most likely form only in the presence of plasminogen, which brings the two molecules into close proximity.

C. Data and Code availability

6. Please include source data files for all data presented in graphs within the figures. Given that the data generated in this study will be very useful to the larger community, in addition to submitting the raw data to ProteomeXchange consortium, the authors should consider making their processed data for proteomics analysis (for instance, normalised log₂ transformed tables as CSV table) and data used to generate the figures easily available. These non-restricted data can be provided as supplementary material or through third party hosting options. Authors should provide a spreadsheet as supplemental data for differential expression comparisons to accompany

proteomics data represented in the figures as it is hard to interpret from plot visualisation alone.

We have compiled all related source data into a single Excel file that contains all necessary processed data, differential analysis results, and other statistical summaries required to reproduce all figures in the manuscript. This information is found in Combined Source Data file.

7. In the interests of data reproducibility, transparency and open science, the authors should also share their scripts for performing the computational analysis related to proteomics experiments shown in this manuscript in a code sharing platform like GitHub/GitLab etc.

We have uploaded our custom R scripts for data processing and visualization related to the proteomics and biochemical experiments to the Zenodo repository, along with examples of input and expected outputs. The files can be accessed using the accession number 13271267 (temporary access link for reviewers). A “Code Availability” section has been added to the manuscript, which includes a source data statement. The scripts will be made publicly available upon publication.

D. Minor comments

line 66: surface dehydrogenase is the same protein as GAPDH

Thanks for pointing this out. The term “surface dehydrogenase” has been changed to “Glyceraldehyde 3-Phosphate Dehydrogenase (GAPDH)” throughout the manuscript.

lines 81-83: Suggest including reference to roles of SLO in translocating NADase into cells (Madden et al. PMID 11163247) and in activating superantigen SSA (Brouwer et al. PMID 33024089).

We have added statements and included the corresponding references regarding the roles of SLO in translocating NADase into cells and in activating the superantigen SSA, as reported by Madden et al. (PMID 11163247) and Brouwer et al. (PMID 33024089). These updates provide a more complete view of SLO's multifunctional capabilities beyond cytolysis.

“Beyond this main biological function, SLO also translocates NAD-glycohydrolase into host cells, promotes streptococcal superantigen activity, acts as an immune-modulatory molecule for neutrophils, and impairs phagocytic clearance of GAS and intracellular lysosomal killing.”

Line 108: Add a phrase justifying why GFP was chosen as a reference control.

We have added a sentence to the manuscript explaining the rationale of using tagged GFP protein as a reference control.

“The tagged GFP bait was used to filter out non-specific protein interactions, determine the baseline background noise, and facilitate comparison with previous studies.”

Line 118: List relevant IgG chains that were specifically enriched by SLO or SCPA.

We have updated the manuscript to include a detailed list of the IgG chains that were uniquely and significantly enriched by the GAS baits, see at **Supplementary Data Table 1**.

bait-prey association	MiST		bait-prey association	MiST
SLO (interacts with) IGHV3-49	0.85465		SCPA (interacts with) IGLV9-49	0.98713
SLO (interacts with) IGHV3-38	0.83959		SCPA (interacts with) IGLV3-9	0.88015
SLO (interacts with) IGKV6D-21	0.80797		SCPA (interacts with) IGKV1-12	0.84374
SLO (interacts with) IGKV2D-28	0.80559		SCPA (interacts with) IGKV1-5	0.80931
SLO (interacts with) IGHV4-61	0.7931		SCPA (interacts with) IGKV4-1	0.79063
SLO (interacts with) IGLL5	0.77996		SCPA (interacts with) IGHV3OR16-9	0.7852
			SCPA (interacts with) IGHV1OR15-1	0.76964
			SCPA (interacts with) IGHV1-3	0.7573

Supplementary Data Table 1

Line 142: The statement that only PLG was significantly enriched in 10% plasma seems to contradict slightly the figures. While PLG is the most significant of the three proteins (of PLG, APOE and CLU), the fold change is lower than for APOE and similar for CLU, and all three proteins are above the log P value threshold of -2.

We realize that the text and the figure were misleading. We have reanalysed these results including the MiST score (PLG: 0.76339, APOE: 0.64791, CLU: 0.54192), the FDR-corrected multiple t-tests, and the two-way ANOVA for multiple comparisons. Collectively, these analyses showed that PLG is the most significant binder across all conditions. Affinity-enrichment mass spectrometry (AE-MS) analysis is associated with some degree of technical variation and can at times be limited by unspecific enrichment or background noise. Consequently, relying on fold-change alone can be misleading. We believe that the combined approach used here ensures a more robust selection of significant binders. We have also clarified this in the manuscript “Results” section on page 7 and “Methods” section on page 19.

Line 328: This sentence requires a reference.

A reference supporting the statement has been added to the manuscript.

Lines 351-2: My understanding is that SKA is encoded by all GAS strains, as it is part of the core genome (see: <https://www.biorxiv.org/content/10.1101/2023.08.29.555273v1.full>).

In our manuscript, we used the definition of the core genome proposed by Davies, M.R., McIntyre, L., Mutreja, A. et al.². According to their work, SKA is not classified as a core gene in GAS. We found additional support for this statement in Le Breton, Y. et al.³, that also excluded SKA from the GAS core genome. We have clarified this information to the manuscript “Discussion” section on page 14.

Lines 352-354: References should be cited to support these statements.

We have added the appropriate references to support the statement concerning the polymorphism of the a1a2 motif of PAM.

E. Methods

Lines 413-416: Please provide details about the protein constructs, including primary sequence and the exact residues included for SLO and SCPA. Both proteins possess an N-terminal signal peptide and SCPA also has a C-terminal sorting signal that is cleaved to form the mature polypeptide.

We have included the requested information in the **Supplementary Notes**. In our study, the N-terminal signal peptide was removed from both GAS baits to reflect their mature forms in our experiments.

Line 476: More details should be provided on parameters used for STRING network. For example, version used, rationale of proteins selected for network (i.e. proteins with differential abundance?), how were the proteins grouped, what active prediction methods were used, interaction scores, etc.

We have updated the “Methods” section to include detailed information used for the STRING interactome and network analysis, on page 19. We have added the version number of the STRING database and the date of access in the editorial reporting summary file as well. The rationale for selecting proteins for network analysis was based on their MiST interaction scores using a threshold of >0.5. The network analysis involved merging a STRING prey-prey link network (generated by querying 68 prey

protein in STRING database) with an SLO-focused MiST bait-prey link subnetwork. Proteins within this network and associated edges were extracted manually into three groups: complement-related proteins, immunoglobulin-related proteins, and other plasma proteins. This information has been added to both “Methods” section and the figure legends.

Line 476-477: Add software versions for Cytoscape, STRING (website), R and Metascape.

We have updated the editorial reporting summary file to include the versions for all software used including Cytoscape (v.3.9), STRING (v.2.0.0, accessed in Apr.2022), R (v.4.2), and Metascape (v.3.5) as requested.

Line 487: Include ethics details of plasma from donor who recently recovered from GAS infection.

We have added ethics details in the manuscript “Methods” section.

“In this study, we used plasma from a donor who recently recovered from a GAS infection. The donor provided written informed consent, and the study was approved by the regional ethics committee (permit number 2015/801). All procedures followed the ethical guidelines outlined in the WMA Declaration of Helsinki and the Department of Health and Human Services Belmont Report.”

F. Figures

Fig 1A: reference domain structure for PLG in the figure legend. Or preferably, include a more detailed figure in the Supplementary showing the domains and amino acid numbering that defines these, and the tPA cleavage site. Also, black font on dark purple circles is hard to see.

We added a schematic figure of PLG in **Supplementary Data Figure 6a**. The Figure illustrates the domains, corresponding residue numbering, and tPA cleavage sites. Additionally, we have updated the colour scheme in the network Figure (see **Figure 1a** for example) to improve text readability and to ensure that all text labels are clearly visible.

Supplementary Data Figure 6a

Figure 1a

Fig 1B: Green circles are not defined in the Key. Also this figure appears to be very similar to Extended Data Fig 3. Are these the same? If so, this should be referenced in the Figure legend. Also it is difficult to distinguish between the colours representing 'regulation of complement activation' and 'reverse cholesterol transport.' Consider formatting to make it clear.

We have clarified the difference between **Figure 1b** and **Supplementary Data Figure 4a** (previously **Extended Data Figure 3**) in the “Results” section and the figure legend. **Figure 1b** is a subset of the full network analysis performed based on 68 SLO-interacting candidates queried in Metascape, which generated enriched terms across various databases including Gene Ontology, Reactome human pathways, and KEGG, etc. The main **Figure 1b** instead only highlights the GO:Biological Process enrichment to make the figure interpretable. **Supplementary Data Figure 4a** shows the complete clustered enriched terms, with additional descriptions and statistical details.

We have changed the colour scheme in **Supplementary Data Figure 4a**, specifically changing the colours of clustered nodes representing “reverse cholesterol transport” .

Supplementary Data Figure 4a

Fig 2A: Is this the 10% diluted plasma? This should be specifically stated in the figure legend.

Yes – that is correct. The figure now specifies that we used 10% diluted plasma. We have updated the annotation in both figure and figure legend.

Fig 2D: I am confused why the pre-coating with SLO gives such a low ELISA value with anti-SLO mAb (as shown in Fig 2C). Wouldn't the primary antibody bind the SLO used for coating? Or is this figure mislabeled?

The primary anti-SLO mAb used in ELISA was diluted to 1:200, which resulted in a low but yet detectable signal. We included a condition where the wells were coated only with 1 ug SLO as a positive control. All corresponding measurement values are now included in the source data file. We have also updated the figure to present direct comparison between initial coating with PLG and PLM.

Fig 2F: the different SLO conditions (quantity or time?) are not annotated. Presumably these are the same as in Fig 2G but should be annotated in this figure also.

Thank you for your observation. We have merged previous panels Fig 2F-G into a single panel (**Figure 2d**) to clarify that the same legend should be referred to for both figures.

Extended Data Fig 5A - Include tPA cleavage site

The tPA cleavage site has now been included in **Supplementary Data Figure 7a** (previously **Extended Data Figure 5A**).

Fig 3A: While I am not familiar with these plots, the APO and SLO-bound plots look very similar to me (as noted by the authors). I assume the differences reflect the magnitude of the Δ DU values in the boxed regions? Also, the amino acid ranges in the text on line 187 don't match the annotations in the figure (e.g. "regions spanning residues 74-84 in PAN" is annotated as "74-88" in Fig 3A). Are these referring to the same thing? Same goes for Lines 192-3 and Extended Data Fig 5B.

Reviewer 2 also pointed this out, which made us realize that previous graphical representation is not optimal. We have now moved the butterfly plot to **Supplementary Data Figure 7b**, and replaced it with a new figure panel in **Figure 3a**. This panel projects the significant HDX-MS peptides onto the PLG structure in surface presentation to better visualize the temporal protein dynamics changes in PLG when bound to SLO. Based on the **Supplementary Data Figure 7c**, the text now accurately reflects the overlapping regions from the 30 s, 300 s, and 3000 s deuteration experiments. For example in the PLG PAN domain, peptides 34-50, 77-88, 34-48, 80-90, 80-88, 79-88, 79-80, and 80-90 are all determined to be significantly protected within 300 s labelling. The core overlapping regions is thus 34-48 and 80-88.

Figure 3a

Extended Fig 1: The authors should comment on the fact that one GPF bait control is different. Is there a reason why this control is different from the others?

There was a mistake in this graph. We have corrected the error and updated the graph. Related R script and the source data file are provided within this manuscript.

Supplementary Data Figure 2: Tang et al

a

Supplementary Data Figure 2

Extended Fig 2A: Black font on dark purple circles is hard to see. Also, according to ref. 36, a MiST score is used to identify the biologically relevant interactions. The referenced considered interactions with a MiST score greater than 0.75 as significant. Please include the value that was used for your analysis.

We have updated the colour scheme in **Supplementary Data Figure 3a** (previously **Extended Data Figure 3A**) to make the text annotations visible. We have also added the threshold for the MiST score. In this analysis, we used for a threshold of > 0.5, slightly lower than the > 0.65 recommended in other studies⁴. We used this threshold to capture a broader range of both direct and indirect interactions associated with the SLO bait protein. We have explained the rationale for this in the “Methods” section on page 18.

Extended Data 5D: include tPA cleavage site

The tPA cleavage site has been added to **Supplementary Data Figure 7e** (previously **Extended Data Figure 5D**).

Fig 5F: Numbering in figure (192-203) doesn't match what is stated in the text (194-203) on line 312.

We have reviewed the text and figure for consistency and updated the manuscript to accurately reflect the numbering of the protected peptide 192-203, as shown in **Figure 5e** (previously **Fig 5F**).

Reviewer #2 (Remarks to the Author):

We appreciate the time and effort of the reviewer in providing feedback on our manuscript.

Reviewer #3 (Remarks to the Author):

In this manuscript, Tang et al use an integrative structural proteomics strategy to interrogate the interaction of SLO with PLG. My expertise is in the field of structural proteomics and my comments below reflect this. Overall, I found some of the data presentation confusing and it was difficult for the reader to understand how the data supported the conclusions made – I am sure this comes down to presentation and the authors should improve this upon revision.

Thank you for your constructive comments and for highlighting the areas in need of improvement regarding data presentation. We have revised the manuscript to improve clarity and ensure that the data presentation supports our conclusions.

- Line 105 – indicate the affinity tag used.

The affinity tag information has been added to the **Supplementary Notes**.

- The authors use statistical analyses to identify significant SLO interactions (line 128). How was significance determined? Did a protein have to pass their thresholds in both analyses SLO/GFP and SLO/SCPA to be significant?

We have clarified this information in the manuscript on page 7. We determined significant SLO-prey interactions by requiring that the prey showed significance in both SLO/GFP and SLO/SCPA comparisons, with an adjusted p-value < 0.01 and a $\log_2(\text{fold change}) > 1$. For strong SLO-binding IgG clones present in undiluted plasma, we adjusted the $\log_2\text{FC}$ threshold from 2 to 1. Additionally, a two-way ANOVA was performed to ensure the robustness of the significance testing.

- Line 144. The authors state their AE-MS identified three binding partners. Whilst their analysis focused on three, im not sure that ONLY three were detected. Could the authors please clarify?

We have changed the text to avoid confusion. Yes, like reviewer stated, we detected 179 proteins in the pull-down samples, excluding common contaminants. But only three proteins (apolipoprotein E, clusterin, and plasminogen) were consistently determined to be significantly enriched after applying the FDR-corrected multiple t-tests and fold change cut-off in both the SLO/GFP and SLO/SCPA comparisons across all three dilution conditions (undiluted, 50% and 10% prey plasma).

“In conclusion, we used AE-MS to map two plasma protein interaction networks formed around SLO or SCPA, and suggest several binding partners, including plasminogen, that specifically interacted with SLO.”

- In Extended Data Figure 2, Figure 1 A, the colours of the nodes should be better explain.

The figure legend has been updated to explain the colour scheme more clearly. In addition to the MiST score, each identified bait-prey interaction comes with three normalized scores representing abundance, specificity, and reproducibility. In the Cytoscape network annotation setting, the node colours now reflect a gradient scale based on specificity scores, ranging from 0.28 to 1. This has been clarified in “Method” section and figure legend.

- What is a MiST score? And why do the width/colour of the edges reflect this? Some explanation is needed in the text/figure legends.

In the revised manuscript, we have added more details to the figure legends and “Methods” section on page 18-19. In addition, we remade the figure so that edge colour in Cytoscape reflects the corresponding MiST score value (also provided within the source data file). The edge width is now uniform across all edges for clarity. The MiST score is calculated based on prey abundance, reproducibility across multiple AE-MS experiments, and specificity compared to other baits^{5,6}.

“Nodes represent proteins, with node size and colour indicating abundance and specificity metrics of the enriched prey protein, while the connecting edges represent protein-protein interactions (PPIs). Edge colour reflects the MiST score value.”

- In Extended Data Figure 1, what statistical test was performed here for proteins to be shown in this heatmap? More information in the legends is needed.

We have added more information to “Methods” section on page 19 and also to the figure legends to explain the data preprocessing/statistical test applied. Briefly, in **Supplementary Data Figure 2a** (previously **Extended Data Figure 1**), we first log₂-transformed the LFQ intensities, filtered out common contaminants, and applied missing data imputation. We then performed row-level z-score normalization before generating the clustered heatmap to visualize the clustering of both protein groups and bait experiments.

- The authors choose to characterise the interaction of plasminogen with SLO by HDX-MS and XL-MS. Is the affinity of the interaction known? In the HDX experiments what % bound plasminogen was used to prepare the samples. This is vital knowledge to correctly design these experiments.

The exact affinity of the interaction between plasminogen (PLG) and SLO is not known. However, our cross-linking experiments indicated that an interaction could be captured at a 1:1 molar ratio with a 2-hour reaction time. While prior knowledge of binding affinity, dissociation constants, and stoichiometry is certainly preferable, it is not uncommon for HDX-MS experiments to be performed without this information.

Without delving too deeply into the technical pros and cons of HDX-MS, it is important to note that this method reveals the average state of the protein in solution at the investigated time point. Specifically, if the unbound form of plasminogen is predominant, no interaction will be observed. Moreover, if 50% of the plasminogen is bound while 50% remains unbound, a false bimodal (EX1) isotope distribution would be observed, which was not the case in our study.

In our HDX-MS experiments, we used an excess molar ratio (2-fold compared to PLG) of SLO, which was sufficient to observe an effect on the dynamics of plasminogen. Increasing the amount of SLO further to maximize the percentage of bound plasminogen would have compromised the quality and coverage of the retrieved plasminogen peptides due to peptide overlap and ion suppression effects.

- Plot in Figure 3A. The authors state in the text (line 187) that minimal differences between states were observed except in three key regions. It is unclear to me how they came to such a conclusion from the data presented. Was a statistical analysis performed? It appears to me that there are other peptides with differences in uptake of comparable levels to these three regions. Why are these three regions indicated? A presentation of these data as a Wood's plot would be easier to interpret, and the statistical analysis could be presented here too. This analysis also seems inconsistent with the data they present in Figure 3B. Why did the authors choose to plot the data for only the PAN and PSD domains in Figure 3E, given that they see other changes in other regions of the protein. It would be beneficial to plot some of these changes onto structures etc of the proteins.

We agree that our previous statement (line 187) was unclear. To address this, we have revised both the graphical representation and the accompanying text. In response to the reviewer's suggestion, we have applied an FDR-controlled significance test for each peptide at a confidence level of 99.9% across different labelling intervals, projecting all identified peptides onto the reference PLG structure. This approach reveals temporal dynamic changes in the protein complex over different deuteration intervals.

Significantly protected and deprotected peptides are now highlighted in the revised **Figure 3A** as below: *“a) Left: annotation of the seven domains in PLG shown in surface presentation. Right: projection of significantly protected/deprotected HDX-MS identified peptides onto the reference structure of PLG. Deprotected, protected and non-significance peptides was determined by peptide-level FDR-controlled significance test using a confidence level of 99.9%, across the different labelling intervals.”*

Figure 3a

The reviewer noted that several non-significant peptides exhibit trends toward decreased dynamics, which we have now clarified in the text: *“In our first analysis, PLG peptides with significantly decreased/increased deuterium uptake were projected onto the PLG structure in surface presentation (Figure 3a), indicating a global stabilization of PLG over time.”*

We also introduced a Woods plot for the 9000 s labeling experiment in **Figure 3b**, with significant peptides colored accordingly, to provide a clearer interpretation of these data. The butterfly plot initially used to profile sum deuterium uptake has been moved to **Supplementary Data Figure 7b**.

The PAN and PSD domains were selected for detailed analysis in the barcode plot for their high sequence coverage, which provides a clearer demonstration of the observed changes. Additionally, the involvement of some of HDX-derived significant protected regions in binding was further corroborated by XL-MS data. In this context, it was important to rely on several techniques, as XL-MS allows us to differentiate between binding surface protection and allosteric stabilization due to the interaction.

- Why do the authors study PLG-SLO and PLM-SLO by XL-MS, but only PLG-SLO by HDX-MS?

Based on our findings from indirect ELISA, XL-MS, and the plasmin activity assays, SLO binding to PLM is not as evident and would thus have a much lower affinity

compared to PLG, making HDX unfeasible. For this reasons, we focused our HDX-MS studies on the PLG-SLO interaction.

- Of the identifications with the different software packages used for XL-MS, how many were unique, and why did the authors only choose to visualise the pLink results? (line 226)

The unique cross-links identified by the different search engines are now summarized in **Supplementary Data Table 2**. We have now also performed a side-by-side comparison of results from both search engines for DSS and DSG cross-linked datasets and added a **Supplementary Data Figure 8a**. We decided to visualize the pLink 2 results in the main Figures as this search engine identified fewer cross-linked peptides that were only supported by a single cross-link spectrum match (CSM).

Supplementary Data Figure 8a

- Only one XL was detected and the authors propose that this is likely because the interaction is weak. How do the authors define weak, without a Kd measurement? Are there Lys residues in/adjacent to protected regions from HDX, that might be crosslinkable, or is this experimental design flawed by the choice of the wrong crosslinking reagents? (or proximal Lys residues in the structural models shown later)

The single cross-linked spectrum match between plasmin and streptolysin O could be due to either transient interactions or potential impurities in the commercially acquired plasmin, which might contain traces of plasminogen. Although we did not perform quantitative XL-MS, other studies have reported that inter-protein cross-linking intensity correlate to some extent with binding affinity (Kd)⁷. To avoid confusion, we have revised the text to state *“The results revealed that SLO binding to PLG is significantly higher compared to PLM.”* rather than “doesn't bind strongly to PLM.”

We can confirm that there are Lysine residues in/adjacent to protected regions from HDX, which are cross-linkable.

- The authors should make it clear what concentrations of crosslinker was used, and show SDS-PAGE gels demonstrating the quantity of crosslinked material and the quality of their experimental design. This should be the case for all crosslinking reactions.

This information, including linker concentration and SDS-PAGE gels from the cross-linked experiments, has been added to **Supplementary Data Table 5** in accordance with community standards.

- The authors state that the ‘fraction of overlength Ca-Ca distances were higher in SLO c.f. PLG (line 252, Figure 4C). From the figure it is hard to justify this statement. It might be more correct that more overlength crosslinks exceed ca 75 Å? Is this what the authors are referring to? If so, how many crosslinks fit this criterion?

Previously, we did not set a strict cut-off for intra-links within the multidomain PLG, as previous studies have reported DSS intra-link distances of up to 59 Å in other multidomain proteins, such as BSA⁸. In our analysis, we found that the fraction of far over-length cross-links (exceeding Ca-Ca distance of 60 Å) was higher in SLO compared to PLG. Specifically, for DSS XL-MS datasets, 3.8% of the cross-links in PLG were over-length, compared to 12.3% in SLO. For DSG XL-MS datasets, the over-length percentages were 3.2% in PLG versus 17.2% in SLO. Comparison has been added to “Results” section on page 11. This suggests that the SLO structure may exhibit more extensive conformational flexibility or variability in solution.

- The authors then interpret these results in the context of SLO dynamics. It is not clear to me how they draw the conclusion made (line 255), without comparing the same data between the apo and bound states.

The dynamic behaviour of SLO during pore-formation is known, like other well-characterised cholesterol-dependent-cytolysin family proteins⁹. Many of the intra-links observed in our study are not consistent with its pore-formation state, indicating that the interaction with PLG induces a unique conformational state in SLO.

- What do the authors mean by targeted crosslinking (line 275)? Is this referring to the computational procedure used? If so, this is misleading. This whole latter section of the results and the way the data are presented in Figure 5 were very difficult to digest and understand. The authors should work to improve the clarity of this complex analysis.

Thank you for pointing this out. We removed the reference to TX-MS from the figure title to avoid confusion and changed the structural figures to better illustrate the

detected cross-links utilized in the computational modelling (**Figure 5**). We have also revised the text describing this Figure. Lastly, we have reworked the supplementary figures to offer additional details regarding the computational analysis.

The TX-MS protocol (targeted cross-linking mass spectrometry) is a workflow¹⁰ developed and named in our laboratory that we have used in several projects and publications^{11,12}. TX-MS features its own MS/MS analysis workflow, that complements state-of-the-art cross-linking tools. Given the complexity of SLO's conformational dynamics, we employed TX-MS alongside conventional cross-linking search tools to ensure accurate statistical results.

Figure 5c

- The authors state that the AlphaFold model could not predict the correct orientation of SLO with respect to PLG or explain the inter-links detected... (line 284). Were the predictions of these regions poor (i.e. pLDDT scores low), or did the model not agree with their data? The authors should show data that would support both of these situations and explain which is true.

In our analysis, we intend to highlight that the AlphaFold (AF) predicted model did not meet the expected standards based on the confidence metrics (pLDDT and pTM), and it also did not align with our experimental data. The scores reported by AF3 (ipTM = 0.15 and pTM = 0.4) indicate a likely failed prediction. Although AF3 shows moderate confidence in each subunit (pLDDT between 70 and 90), it reported a low score for the binding interface (pLDDT < 50), underscoring the limitations of the model (as shown in **Figure R1**). Furthermore, the model fails to corroborate our experimental data, particularly the interprotein cross-links. These unsupported inter-links are detailed in **Supplementary Data Table 3**.

Figure R1: AF3 Prediction Results for the PLG-SLO Complex. The metrics are displayed at the top. The structure is color-coded according to the pLDDT scores.

- Line 297. How many structures were consistent with most crosslinks? Was this a rare event? How many crosslinks were not satisfied? Were these the 4 crosslinks that were consistent with the alternate conformer proposed in Figure 5E?

We have added the number of structures consistent with the most numbers of interprotein cross-links to **Supplementary Data Figure 12a**. When selecting the top model, we considered both the docking score, which represents stability, and the number of consistent cross-links. Top model and alternative model were consistent with different but overlapped set of cross-links as detailed in **Supplementary Data Table 3**.

Supplementary Data Figure 12a

- Figure 5F – not clear how this relates to the statement on line 311. Surely it would be better to do a side-by-side comparison of HDX data with the predicted interface.

We agree with this comment and have now added a side-by-side comparison between HDX data and the predicted interface in **Figure 5e** (previously **Figure 5F**).

Figure 5e

- Community standards for reporting HDX [Nature Methods, 16, 595–602 (2019)] and XL [Anal. Chem. 91, 6953–6961 (2019)] data should be followed.

Both community-standard HDX-MS and XL-MS data summary now can be found in **Supplementary Data Table 4-5**.

Reference

1. Wang, X., Lin, X., Loy, J. A., Tang, J. & Zhang, X. C. Crystal Structure of the Catalytic Domain of Human Plasmin Complexed with Streptokinase. *Science* **281**, 1662–1665 (1998).
2. Davies, M. R. *et al.* Atlas of group A streptococcal vaccine candidates compiled using large-scale comparative genomics. *Nat Genet* **51**, 1035–1043 (2019).
3. Breton, Y. L. *et al.* Essential Genes in the Core Genome of the Human Pathogen *Streptococcus pyogenes*. *Sci. Rep.* **5**, 9838 (2015).
4. Verschueren, E. *et al.* Scoring Large-Scale Affinity Purification Mass Spectrometry Datasets with MiST. *Curr Protoc Bioinform* **49**, 8.19.1-8.19.16 (2015).

5. Jäger, S. *et al.* Global landscape of HIV–human protein complexes. *Nature* **481**, 365–370 (2012).
6. Morris, J. H. *et al.* Affinity purification–mass spectrometry and network analysis to understand protein–protein interactions. *Nat Protoc* **9**, 2539–2554 (2014).
7. Hagemann, G. *et al.* Quantitative crosslinking and mass spectrometry determine binding interfaces and affinities mediating kinetochore stabilization. *bioRxiv* 2022.03.31.486303 (2022) doi:10.1101/2022.03.31.486303.
8. Iacobucci, C. *et al.* First Community-Wide, Comparative Cross-Linking Mass Spectrometry Study. *Anal. Chem.* **91**, 6953–6961 (2019).
9. Pee, K. van *et al.* CryoEM structures of membrane pore and prepore complex reveal cytolytic mechanism of Pneumolysin. *Elife* **6**, e23644 (2017).
10. Hauri, S. *et al.* Rapid determination of quaternary protein structures in complex biological samples. *Nat Commun* **10**, 192 (2019).
11. Bahnan, W. *et al.* A human monoclonal antibody bivalently binding two different epitopes in streptococcal M protein mediates immune function. *Embo Mol Med* (2022) doi:10.15252/emmm.202216208.
12. Happonen, L. *et al.* A quantitative *Streptococcus pyogenes*–human protein–protein interaction map reveals localization of opsonizing antibodies. *Nat Commun* **10**, 2727 (2019).

Response to Reviewer comments

Dear reviewers,

We sincerely thank the reviewers for their recognition of our revised manuscript. We have addressed the reference formatting issues raised by Reviewer #1. Once again, we greatly appreciate all the reviewers' efforts and constructive comments during the review process.

Kind regards,

Di Tang and Johan Malmström on behalf of all authors